# MOMENT CONSTRAINED OPTIMAL TRANSPORT FOR CONTROL APPLICATIONS

## ABSTRACT

This paper concerns the application of techniques from optimal transport (OT) to mean field control, in which the probability measures of interest in OT correspond to empirical distributions associated with a large collection of controlled agents. The control objective of interest motivates a one-sided relaxation of OT, in which the first marginal is fixed and the second marginal is constrained to a "moment class": a set of probability measures defined by generalized moment constraints. This relaxation is particularly interesting for control problems as it enables the coordination of agents without the need to know the desired distribution beforehand. The inclusion of an entropic regularizer is motivated by both computational considerations, and also to impose hard constraints on agent behavior. A computational approach inspired by the Sinkhorn algorithm is proposed to solve this problem. This new approach to distributed control is illustrated with an application of charging a fleet of electric vehicles while satisfying grid constraints. An online version is proposed and applied in a case study on the ElaadNL dataset containing 10,000 EV charging transactions in the Netherlands. This empirical validation demonstrates the effectiveness of the proposed approach to optimizing flexibility while respecting grid constraints.

## 1 INTRODUCTION

**Optimal Transport** Optimal Transport (OT) theory first emerged in the 18th century, and more recently has become a significant tool in the machine learning toolbox (Villani, 2008; Peyré et al., 2019). The goal is simply described: given two random variables $X$ and $Y$, find a joint probability measure $\pi^*$ for the pair $(X, Y)$ that preserves the marginals, and minimizes some criterion. When $X$ and $Y$ belong to a common state $\mathcal{X}$, the Monge-Kantorovich formulation is expressed as follows.

Let $\mathcal{U}(\mu_1, \mu_2) = \{\pi \in \mathcal{B}(\mathcal{X} \times \mathcal{X}) \colon \pi_1 = \mu_1, \ \pi_2 = \mu_2\}$ where $\pi_i$ denotes the $i$th marginal, for example $\pi_1(dx) = \int_{\mathcal{X}} \pi(dx, dy)$, and with $\mathcal{B}(\mathcal{X} \times \mathcal{X})$, the set of Borel probability measures on $\mathcal{X} \times \mathcal{X}$. Given a cost function $c \colon \mathcal{X} \times \mathcal{X} \to \mathbb{R}_+$, the optimal transport problem is formulated as the minimum

$$\min_{\pi} \left\{ \int_{\mathcal{X} \times \mathcal{X}} c(x, y)\pi(dx, dy) \ : \ \pi \in \mathcal{U}(\mu_1, \mu_2) \right\}. \tag{1}$$

Several authors have proposed relaxations of the OT problem, such as *unbalanced OT* where an entropic penalization of the deviation from the marginals is introduced (Chizat et al., 2017). Relaxations of marginals have been considered to improve numerical performance or to approximate the OT problem (Balaji et al., 2020; Le et al., 2021; Alfonsi et al., 2020) but, to the best of our knowledge, never as a natural representation of a Mean Field control (MFC) problem.

**Mean field control** Many academic communities are interested in transforming probability measures efficiently. Examples include the *fully probabilistic control design* of Kárný (1996) and the related linearly-solvable Markov decision framework (Todorov, 2007). The area of mean field games begins with a multi-objective control problem, but the final solution technique amounts to transporting a probability measure on a high dimensional space in such a way as to minimize some objective function. Similar to mean field games is the cooperative setting of *mean field control* or *ensemble control*, with applications (Hochberg et al., 2006; Chertkov & Chernyak, 2018) ranging from power

systems to medicine; This technique can also be relaxed (Cammardella et al., 2020; Bušić & Meyn, 2018). More examples may be found in the survey of Garrabe & Russo (2022).

We are interested in the following control problem. Consider a set of $K$ agents, whose *state* is denoted $X_k = (S_k, W_k) \in \mathcal{X}$ for each $1 \leq k \leq K$. It is assumed that $S_k$ is an *exogeneous variable*, while $W_k$ is fully controllable. Given a cost function $c \colon \mathcal{X} \to \mathbb{R}$ and a constraint function $f \colon \mathcal{X} \to \mathbb{R}^M$, we seek to minimize:

$$\min_{W_k}\Big\{\sum_{k=1}^{K} c(X_k) \ : \ \sum_{k=1}^{K} f(X_k) \leq 0\Big\} \tag{2}$$

This general formulation allows for control of dynamical systems, in which case the state space $\mathcal{X}$ is the set of possible sample paths. The optimization problem is designed for distributed control applications in which the global constraint is interpreted as coordinating the ensemble of agents, and the cost $c$ represents a penalty for deviation from nominal behavior, as is the case in Chertkov & Chernyak (2018); Cammardella et al. (2020); Bušić & Meyn (2018).

The mean field limit of this problem corresponds intuitively to $K \to \infty$:

$$\min_{\mu}\Big\{\int_{\mathcal{X}} c(x)d\mu(x) \ : \ \int_{\mathcal{X}} f(x)d\mu(x) \leq 0 \ \text{and} \ \mu_1 = \nu\Big\} \tag{3}$$

in which $\mu$ is the distribution of $X = (S, W)$, and $\nu$ is the first marginal of $\mu$—the distribution of the exogenous variable $S$. It is important to note that the optimization is only done on the control variable (e.g. plugging time of an EV) and the distribution $\nu$ (e.g. distribution of the arriving time and battery level of an EV) is not modified; this is what we will subsequently call "preserving the distribution of the exogenous variables".

Often in the Mean Field litterature, a Kullback-Leibler cost term is introduced as a regularizer (Chertkov & Chernyak, 2018; Todorov, 2007) and similar control objectives, but with the constraints on the functions $f$ relaxed through a quadratic penalty have been addressed (Cammardella et al., 2020; Bušić & Meyn, 2018). Inspired by the similarities between the OT problem (1) and the Mean Field Control applications such as (3), we want to build bridges between these fields and investigate how computational techniques from OT theory might apply to the computation of optimal control solutions.

**Contributions** We introduce *Moment Constrained Optimal Transport for Control* (MCOT-C) which is a natural representation of a MFC problem designed to achieve three objectives:

- Coordination of an ensemble of agents to achieve a desired goal.

- Enforcement of physical constraints, both spatial and dynamics.

- Enforcement of strict constraints on the distribution of exogenous variables.

Instead of considering the whole state space often very large or even infinite dimensional (e.g. trajectories of agents), this approach focuses on a finite set of moments, relevant to the control objective (e.g. signal tracking). This leads to a tractable algorithm: we modify the Sinkhorn algorithm (Cuturi, 2013) by replacing the update on the second marginal by gradient descent on the dual. An MFC application on charging a fleet of electric vehicles (EVs) while satisfying grid constraints is used to illustrate this new approach. This MCOT-C setting is then extended in two ways: (i) by an online approach which allows to consider real datasets where the algorithm discovers at each step the state of the agents, as presented in section 4 with the ElaadNL dataset (OpenDataset, 2019) (ii) by the use of Monte Carlo type methods, which allow tackling MFC problems where the state space is infinite-dimensional, as in the case study on water heaters presented in appendix E.

**Notations** The state space $\mathcal{X}$ is assumed to be a closed subset of $\mathbb{R}^N$ with $N \geq 1$. It is always assumed that $c(x, x) = 0$ for each $x$. For $\pi$ a bivariate distribution on $\mathcal{X}$, its marginals will be denoted $\pi_1$ and $\pi_2$ such that $\forall x \in \mathcal{X}, \pi_1(dx) = \int_{\mathcal{X}} \pi(dx, dy)$ and $\forall y \in \mathcal{X}, \pi_2(dy) = \int_{\mathcal{X}} \pi(dx, dy)$.

Solutions to each problem problem considered will involve a family of probability kernels $\{T^\lambda : \lambda \in \mathbb{R}_+^M\}$. For each $\lambda$ we define $\pi^\lambda$ by $\pi^\lambda(dx, dy) = \mu_1(dx)T^\lambda(dx, dy)$, and let $\mu^\lambda = \pi_2^\lambda$ denote the second marginal:

$$\mu^\lambda(A) := \int \mu_1(dx)T^\lambda(x, A), \qquad A \in \mathcal{B}(\mathcal{X})$$

For measurable $g\colon \mathcal{X} \to \mathbb{R}$ and $f\colon \mathcal{X} \times \mathcal{X} \to \mathbb{R}$, we adopt the operator-theoretic notation,

$$T^\lambda g\,(x) := \int T^\lambda(x, dy)g(y)\,, \ \ \forall x \in \mathcal{X}\,, \qquad \langle \pi, f \rangle := \int_{\mathcal{X} \times \mathcal{X}} f(x, y)\pi(dx, dy)$$

## 2 MOMENT CONSTRAINED OPTIMAL TRANSPORT FOR CONTROL

### 2.1 STATEMENT OF THE PROBLEM

The $m$ components $\{f^m : 1 \le m \le M\}$ of the function $f\colon \mathcal{X} \to \mathbb{R}^M$ define the *moment class*,

$$\mathcal{P}_f = \{\mu \in \mathcal{B}(\mathcal{X}) : \langle \mu, f^m \rangle \le 0 \ : \ 1 \le m \le M\} \tag{4}$$

The equality constraint $\langle \mu, f^m \rangle = 0$ can be expressed as a pair of inequality constraints, so it is possible to impose equality constraints when needed. Recall that for MFC, any probability measure $\pi$ on $\mathcal{B}(\mathcal{X} \times \mathcal{X})$ is subject to the constraint that its first marginal $\mu_1$ is given, and the distribution $\nu$ of the exogenous variable is also fixed. Equivalently, the bivariate distribution $\pi$ belongs to

$$K(\mu_1, \mu) = \{\pi \in \mathcal{U}(\mu_1, \mu) : \pi((x_s, x_w), (y_s, y_w)) = \mu_1(dx_s, dx_w)T((x_s, x_w), dy_w)\delta_{x_s}(dy_s)\}$$

where $\delta$ the Kronecker symbol, and $T$ ranges over all probability kernels. That is, if $\pi \in K(\mu_1, \mu)$, then $\int_{\mathcal{W}} \pi_2(y_s, dy_w) = \int_{\mathcal{W}} \pi_1(y_s, dx_w) = \nu(y_s)$, which corresponds to our objective of preserving $\nu$ on $\mathcal{S}$. Lastly, we will use the following Kullback Leibler (KL) regularizer:

$$D_{\mathrm{KL}}(\pi \| \mu_1 \otimes \mu_2) = \int_{\mathcal{X} \times \mathcal{X}} \log\left(\frac{\pi(x, y)}{\mu_1(x)\mu_2(y)}\right)\pi(dx, dy) \tag{5}$$

The probability measure $\mu_2$ in 5 may be chosen based on intuition regarding the form of $\pi_2^*$, chosen for ease of computation, or designed to encode hard constraints.

This allows us to introduce the Mean Field Control problem:

**Problem MCOT-C:** *Moment Constrained Optimal Transport for Control*

$$\min_{\pi, \mu}\{\langle \pi, c \rangle + \varepsilon D_{\mathrm{KL}}(\pi \| \mu_1 \otimes \mu_2) : \pi \in K(\mu_1, \mu)\,, \ \mu \in \mathcal{P}_f\} \tag{6}$$

### 2.2 DUAL PROBLEM

This subsection defines the dual and the theoretical properties needed for the algorithm but more details on duality theory and proofs may be found in the appendices A and B. The theoretical results of this problem in the Gaussian case are presented in appendix C. An example that illustrates the impact of regularization can be found in appendix D.

**Assumptions** Assumptions are required for the existence of optimizers and desirable properties of the dual:

**(A1)** $c\colon \mathcal{X} \times \mathcal{X} \to \mathbb{R}_+$ and $f\colon \mathcal{X} \to \mathbb{R}^M$ are continuous, and there is an open neighborhood $N \subset \mathbf{R}^M$ containing 0 such that $\mathcal{P}_{f-r}$ is non-empty for all $r \in N$.

**(A2)** $\mu_1$ and $\mu_2$ have compact support, and the problem is feasible under perturbations: for any $r \in N$, there is $\pi$ and $\mu$ satisfying $\mu \in \mathcal{P}_{f-r}$ and $\pi \in \mathcal{U}(\mu_1, \mu)$.

**(A3)** $\Sigma^0 := \mathrm{Cov}\,(Y)$ is positive definite when $Y \sim \mu_2$.

**Dual** The dual of MCOT-C is by definition the function $\varphi^*\colon \mathbb{R}_+^M \to \mathbb{R} \cup \{-\infty\}$,

$$\varphi^*(\lambda) = \varepsilon \min_{\pi, \mu}\{-\langle \pi, l \rangle + D_{\mathrm{KL}}(\pi \| \mu_1 \otimes \mu_2) : \pi \in K(\mu_1, \mu)\} \tag{7}$$

For each $\lambda \in \mathbb{R}_+^M$, $\varepsilon > 0$ and $x = (x_s, x_w) \in \mathcal{X}$, we denote

$$B_{\lambda, \varepsilon}(x) = \varepsilon \log \int_{y_w \in \mathcal{W}} \exp\big(\varepsilon^{-1}(\lambda^\intercal f((x_s, y_w)) - c((x_s, x_w), (x_s, y_w)))\big)\mu_2(dy_w) \tag{8}$$

**Proposition 1.** *Subject to (A1)–(A3),*

**(i)** *The infimum* (7) *gives* $\varphi^*(\lambda) = -\langle \mu_1, B_{\lambda,\varepsilon} \rangle$.

**(ii)** *The maximizer is* $\pi^\lambda(dx, dy) = T^\lambda(x, dy)\mu_1(dx)$ *with*

$$T^\lambda(x, dy) = \mu_2(dy)\exp(L^\lambda(x,y)), \qquad L^\lambda(x,y) = \varepsilon^{-1}\{\lambda^\intercal f(y) - c(x,y) - B_{\lambda,\varepsilon}(x)\}, \quad (9a)$$

*and* $\mu^\lambda(y) = \pi_2^\lambda(y) \quad \forall y \in \mathcal{X}$

**(iii)** *There is no duality gap: there is a unique* $\lambda^* \in \mathbb{R}_+^M$ *satisfying*

$$\varphi^*(\lambda^*) = \min_{\pi,\mu}\left\{\langle\pi, c\rangle + \varepsilon D_{KL}(\pi\|\mu_1 \otimes \mu_2) : \pi \in K(\mu_1, \mu), \; \mu \in \mathcal{P}_f\right\} \qquad (9b)$$

It is convenient to make the change of variables $\zeta = \varepsilon^{-1}\lambda$, and consider

$$\mathcal{J}(\zeta) := -\varepsilon^{-1}\varphi^*(\varepsilon\zeta)$$

We turn next to the representation of the derivatives of the dual function. The quantity $\varepsilon^{-1}B_{\varepsilon\zeta,\varepsilon}(x)$ is a log moment generating function for each $x$; for this reason, it is not difficult to obtain suggestive expressions for the first and second derivatives with respect to $\zeta$.

**Proposition 2.** *The function* $\mathcal{J}$ *is convex and continuously differentiable. The first and second derivatives of* $\mathcal{J}$ *admit the following representations:*

$$\nabla\mathcal{J}(\zeta) = m^\lambda, \qquad \nabla^2\mathcal{J}(\zeta) = \Sigma^\lambda \qquad (10a)$$

*in which* $m_i^\lambda = \langle\mu^\lambda, f_i\rangle = \mathsf{E}^\lambda[f_i(Y)]$ *for each i, and the Hessian* (10a) *coincides with the conditional covariance:*

$$\Sigma^\lambda = \mathsf{E}^\lambda[f(Y)f(Y)^\intercal] - \mathsf{E}^\lambda[\mathsf{E}^\lambda[f(Y) \mid X]\mathsf{E}^\lambda[f(Y) \mid X]^\intercal] \qquad (10b)$$

It follows that $\mathcal{J}$ is strictly convex:

**Lemma 1.** *Suppose that (A1)–(A3) hold. Then, the covariance* $\Sigma^\lambda$ *is full rank for any* $\lambda \in \mathbb{R}_+^M$.

### 2.3 Algorithm: Semi-Sinkhorn with Gradient Descent

For numerical experiments, the state space $\mathcal{X}$ will be discretized and we will denote by $N$ its cardinality. The cost will be represented by a matrix $C \in \mathbb{R}_+^{N \times N}$. The solution to MCOT-C obtained in Proposition 1 may be expressed

$$\pi_{i,j}^* = u_i K_{i,j}\exp\left(\zeta^{*\intercal}f_j\right) \qquad (11)$$

where $K$ is the Gibbs kernel defined by $K_{i,j} = \exp(-C_{i,j}/\varepsilon)\mu_{2,j}$ and $u_i = \mu_{1,i}/\sum_j C_{i,j}e^{\zeta^{*\intercal}f}$. As shown in Proposition 2, it is possible to obtain a gradient descent algorithm, which looks similar to the Sinkhorn Algorithm (Cuturi, 2013), the difference being the update of $\zeta^k$.

---
**Algorithm 1** Semi-Sinkhorn with Gradient Descent

**Input:** $\mu_1, C, f$
$\zeta^0 \leftarrow \mathbf{0_M}$
$k \leftarrow 0$
**while** $k \le Kmax$ **do**
  $u_i^{k+1} \leftarrow \mu_{1,i}/\sum_j C_{i,j}e^{\zeta_k{}^\intercal f}$
  $\zeta^{k+1} \leftarrow \zeta^k + \sum_{i,j} f_j u_i^k C_{i,j}e^{\zeta^k{}^\intercal f}$
  $\zeta^{k+1} \leftarrow \max\{0, \zeta^{k+1}\}$
  $k \leftarrow k + 1$
**end while**

---

It is also possible to perform Newton's method rather than gradient descent by changing the update of $\zeta_k$ by

$$\Sigma^k \leftarrow \sum_{i,j} f_j f_j^\intercal u_i^k C_{i,j}e^{\zeta_k{}^\intercal f}$$

$$\zeta^{k+1} \leftarrow \zeta^k + (\Sigma^k)^{-1}\sum_{i,j} f_j u_i^k C_{i,j}e^{\zeta_k{}^\intercal f}$$

In cases where the starting point $\zeta^0$ is close to the optimum $\zeta^*$, we can obtain quadratic convergence (C.T.Kelley, 1999).

## 3 USE CASE: EV CHARGING

### 3.1 PRESENTATION OF THE USE CASE

Consider a large fleet of electric vehicles (EVs) arriving to a charging station at random times and with random state of charge, according to an initial law $\nu_0$. There is a central planner whose goal is to maintain constraints for the aggregate power consumption, as well as constraints for each vehicle owner. The vehicles arrive during the period $[9\text{am}, 10:30\text{am}]$, and must be fully charged by 5pm.

The goal is power tracking: total power consumption should follow a reference signal $(r_t)$ over a time period $[t_1, t_2]$, with $9\text{am} \leq t_1 < t_2 \leq 5\text{pm}$. This can be formulated as an MCOT-C problem over the space of distributions on $\mathcal{X} = \mathcal{S} \times \mathcal{W}$ with $\mathcal{S} = [0, T] \times [0, 1]$ and $\mathcal{W} = [0, T]$.

The two first coordinates of $x \in \mathcal{X}$ are the time and the battery state of charge at the arrival and the third is the time when the EV will start charging, called the *plugging time*; so $x \in \mathcal{X}$ is of the form $x = (t_a, b, t_c)$. At each iteration, a gradient is calculated on $\mathcal{X} \times \mathcal{W}$, with complexity of $N_t^3 \times N_b$, with $N_t = 25$ and $N_b = 20$, being the number of discretization points in time and battery state of charge. In this example, this value remains relatively low so that Monte Carlo methods (presented in the appendix E) are not required. We use the MCOT-C problem presented in Section 2 with $\varepsilon = 0.03$ being a compromise between computational stability and having a low value (as any non-negative value will enforce the physical constraints). We consider a version of problem MCOT-C with $\mu_1$ modeling the naive decision rule in which a vehicle initiates charging on arrival:

$$\mu_1(t_a, b, t_c) = \left\{ \begin{array}{l} \nu(t_a, b) \text{ if } t_a = t_c \\ 0 \text{ otherwise} \end{array} \right.$$

Initiation of charging must be after the arrival time (physical constraint) and every vehicle must be fully charged no later than 5pm (quality of service constraint). The following distribution meets these requirements, $\mu_2(t_a, b, t_c) = \mathbf{Unif}_{[t_a, T - \frac{1-b}{v}]}(t_c)$, with $v$ being the charging speed and $\mathbf{Unif}_{[a,b]}$ being the density of uniform distribution over $[a, b]$. It is assumed that drivers wish to initiate charging as soon as possible: this makes it easier for the driver to manage an unforeseen event and may make it easier for the central planner to respond to a grid contingency. This preference is modeled through the cost $c((., ., t_c^x), (., ., t_c^y)) = (t_c^x - t_c^y)^2$.

### 3.2 NUMERICAL RESULTS

**EV charging without unplugging** The first results described here impose an additional constraint: once charging begins, it cannot be interrupted until the vehicle is fully charged. In the following simulations, a constraint on power consumption is imposed for the time period beginning at $t_1 = 10\text{am}$ and ending at $t_2 = 12\text{pm}$. As the optimizer $\mu^*$ will be mutually absolutely continuous with respect to $\mu_2$, both physical constraints and constraints on quality of service are imposed through choice of $\mu_2$.

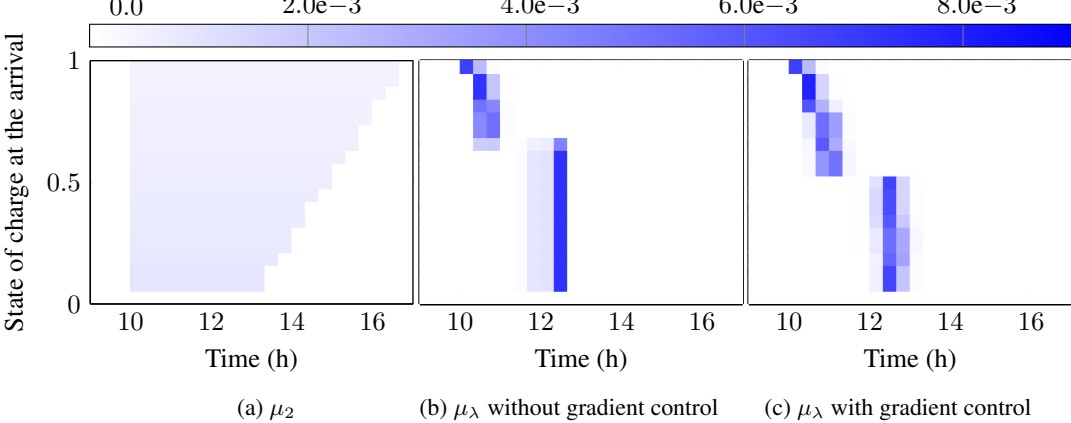

(a) $\mu_2$      (b) $\mu_\lambda$ without gradient control      (c) $\mu_\lambda$ with gradient control

Figure 1: For vehicles arriving at 10am : (a) $\mu_2$ designed to encode physical and quality of service constraints; (b) optimized $\mu$ without gradient control; (c) optimized $\mu$ with gradient control.

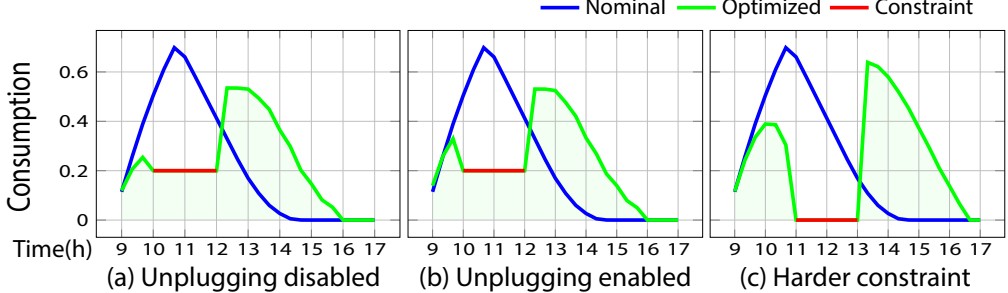

Figure 2: (a) optimized consumption compared to the nominal with unplugging disabled; (b) optimized consumption with unplugging enabled; (c) optimal consumption with constraint infeasible without unplugging.

In Figure 1(a), the constraints enforced on $\mu_2$ can be observed:

- Quality of Service constraint: At 5 pm, all EVs must be fully charged. Thus, if a vehicle needs $\Delta t$ minutes to charge, then the probability of connecting between $5\text{pm}-\Delta t$ and 5pm is zero. This is observed by the completely white lower right triangle.

- Physical constraint: Vehicles cannot load before arriving, so there is no mass probability before 10am for vehicles arriving at 10am.

These constraints are found in the $\mu_\lambda$ showed in Figure 1(b) and 1(c), as $\mu_\lambda$ is a reweighting of $\mu_2$. Aggregated consumption displayed in Fig. 2 (a) shows that the first vehicles to arrive will start charging, but most of those arriving just before 10:00 am will initiate charging only if they arrive with a high battery level so that they are fully charged before the start of the constraint window from 10:00 am to 12:00 pm.

**EV charging with unplugging** The model can be extended by authorizing a vehicle to interrupt and restart charging. In this case, $\mathcal{X}$ is extended with two extra time dimensions corresponding to an unplugging time and a re-plugging time. A second term is included in $c$ that is quadratic in the difference of these times, designed to discourage charging interruption.

We find that unplugging does not impact significantly the optimal solution. Fig. 2 (a) and (b) provide a comparison. Only a slight difference is visible before 10 am: A number of vehicles start to charge before the constraint, stop at 10pm and restart afterwards. However, in some cases, this extra flexibility in charging is necessary to obtain a feasible solution. Fig. 2 (c) shows results obtained when power consumption is not permitted in the middle of the day. In any feasible solution, a portion of vehicles stop charging for a period before they are fully charged.

**Gradient control to flatten the curve** For real-life applications, controlling overall consumption over part of the day through equality of consumption to a predefined signal can lead to a peak when the constraint is released. This phenomenon, due to the penalization of distant charging times, is observed in the different plots of Fig. 2. Consumption can be smoothed by introducing the derivative constraints

$$\forall t \in [0, T], |\langle g_t, \mu \rangle| \leq g_{\max}$$

where $g_t = f_{t+1} - f_t$. In this example, $g_{\max} = 0.2$, thus the overall consumption must not increase by more than 0.2 per hour, which is what we observe in Fig. 3: consumption at 12pm increases more slowly. We can also see the impact of the constraint on the gradient by looking at the difference between Figure 1(b) and 1(c). In both

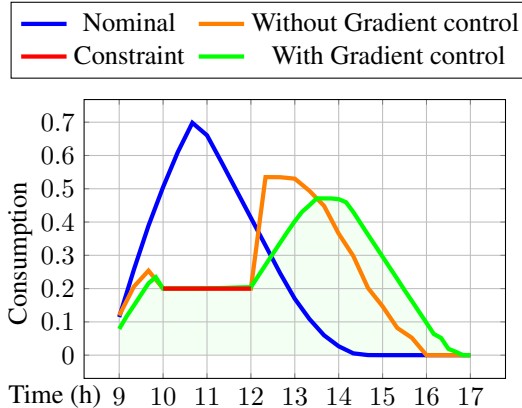

Figure 3: Optimal consumption with and without gradient control of the overall consumption

cases, vehicles arriving with a high battery level are put to charge first. This comes from the quadratic penalty on the start of the charging time: We prefer to charge those which will quickly be completely charged and which will free up space for those which will take longer.

# 4 ONLINE MCOT-C FOR EV CHARGING

In this section, we provide an online version of MCOT-C and test it on a real dataset.

## 4.1 FORMULATION OF ONLINE MCOT-C

First, while some theoretical models assume perfect knowledge of the battery level at each time step (Séguret, 2023), this value is hard to obtain in practice even if estimates are available (Rezvanizaniani et al., 2014) and existing datasets do not take this data into account (Amara-Ouali et al., 2021). Our choice on this subject is to focus on the leaving time $t_l$ and the charging need $\Delta t_n$, which is the charging time requested by the EV owner. These parameters are easier to access and are consistent with other articles studying real datasets (He et al., 2012; Sadeghianpourhamami et al., 2018). Arriving EVs are therefore defined on the following state space:

$$\mathcal{S} = \underbrace{[0,24]}_{\substack{\text{Arriving time} \\ t_a}} \times \underbrace{[0,24]}_{\substack{\text{Leaving time} \\ t_l}} \times \underbrace{[0,24]}_{\substack{\text{Charging need} \\ \Delta t_n}} \times \underbrace{\{0,N_p\}}_{\substack{\text{Max power} \\ p_{max}}} \tag{12a}$$

At each time step $t \in [0,24]$, EVs are controlled through their charging starting time $t_c$. The control space is thus defined as:

$$\mathcal{W}^{(t)} = \underbrace{[t,24]}_{\text{Plugging time } t_c} \tag{12b}$$

and we define the product space: $\mathcal{X}^{(t)} = \mathcal{S} \times \mathcal{W}^{(t)}$. At each time step $t \in [0,24]$, this sequence of actions will take place:

1. New EVs arrive at the charging station and are added to the list of vehicles already present and not charging yet $\{S_i^{(t)}\} = \{S_i : t_a^i \leq t \text{ and } t_c^i \geq t\}$. The empirical $\nu^{(t)}$ is updated:

$$\nu^{(t)}(s) = \begin{cases} \frac{1}{N_t} \sum_i \delta(s - S_i^{(t)}) & \text{if } t_a \leq t \\ \frac{N}{N_t} \nu(s) & \text{if } t_a > t \end{cases} \tag{13a}$$

   where $N_t = \int_{\mathcal{S}} \sum_i \delta(s - S_i^{(t)}) ds + N \int_{\mathcal{S}} \nu(s) \mathbf{1}_{t_a > t}(s) ds$ is the number of vehicles already arrived and not charging plus the number of vehicles that are estimated to arrive.

2. $\mu_1^{(t)}$ is defined by the "Plug when Arrive" strategy: $\forall s = (t_a, t_l, \Delta t_n, p) \in \mathcal{S}$,

$$\mu_1^{(t)}(s, t_c) = \nu^{(t)}(s) \delta(t_c - t_a) \tag{13b}$$

3. $\mu_2^{(t)}$ is defined as "Plug with a uniform distribution" strategy:
   $\forall s = (t_a, t_l, \Delta t_n, p) \in \mathcal{S}, t_c \in \mathcal{W}$,

$$\mu_2^{(t)}(s, t_c) = \begin{cases} \mathbf{Unif}_{[t_a, t_l - \Delta t_n]}(t_c) \nu^{(t)}(s) & \text{if } t_a > t \\ \mathbf{Unif}_{[t, t_l - \Delta t_n]}(t_c) \nu^{(t)}(s) & \text{if } t_a \leq t \end{cases} \tag{13c}$$

   where $\mathbf{Unif}[a,b]$ is the density of the uniform distribution on the segment $[a,b]$. For the sake of simplicity, we assume that there is no outlier (no vehicle that would require more charging time than the difference between their arrival time and leaving time in particular). As in Section 3, $\mu_2$ is designed to incorporate the strong constraint of respecting the quality of service through the absolute continuity of $\mu$ with $\mu_2$ (due to the KL term).

4. The central planner will minimize Equation (6) to obtain:

$$\pi^{(t)} = \underset{\substack{\pi \in K(\mu_1^{(t)}, \mu) \\ \mu \in \mathcal{P}_{f(t)}}}{\arg\min} \; \langle \pi, c \rangle + \varepsilon D_{\mathrm{KL}}(\pi || \mu_1^{(t)} \otimes \mu_2^{(t)})$$

The function $c$ chosen here is a quadratic penalization: $c((s^x, t_c^x), (s^y, t_c^y)) = (t_c^x - t_c^y)^2$. In this case, as we compare it with the "Plug When Arrive" strategy for which $t_c^x = t_a^x$, $c$ is a penalty for starting charging long after the vehicle arrives.

5. For each vehicle $S_i^{(t)}$, its plugging time $t_c^i$ is randomly chosen according to $\pi_2^{(t)}(S_i^{(t)}, .)$. $f$ is then updated as: $f^{(t+1)} = f^{(t)} + \frac{1}{N} \sum_{t_c^i = t} f(S_i^{(t)})$. Vehicles $S_i^{(t)}$ such that $t_c^i = t$ begin their charging.

## 4.2 ALGORITHM

In Algorithm 2, $Alg(\zeta^{(t)}, \mu_1, \mu_2)$ returns $\zeta^{(t+1)}$ the value of Algorithm 1 with the stopping criterion $N_t \|(\langle f^{(t)}, \mu_{\zeta^{(t)}} \rangle)^+\| \leq N\kappa$ and $(.)^+$ is the positive part function: $\forall x \in \mathbb{R}^M, (x)_m^+ = \max(0, x_m)$. The norm $\|\|$ can be chosen as desired but a good candidate is the infinite norm. In general, $\kappa$ is chosen relatively small, and with this norm, $N\kappa$ corresponds to the maximum error on all the vehicles that we can afford to have, we can estimate that this error evolves linearly with N, which explains the multiplication by $N$ (it is important to remember that N is the order of magnitude of the vehicles that will arrive during the day). We define the convergence error at time $t$ as $\mathcal{E}_t(\zeta) = \frac{N_t}{N} \|(\langle f^{(t)}, \mu_{\zeta^{(t)}} \rangle)^+\|$ and $\nu_r$, the real arrival law of EVs. With the definitions of $\mu_2^{(t)}$ and $\mu_1^{(t)}$ in Equations (13) and Proposition 1, we define $F_\zeta$ as: $\forall s \in \mathcal{S}, F_\zeta(s) =$

$$\begin{cases} \dfrac{\int_{\mathcal{W}} \mu_\zeta^{(t)}(s, t_c) f(s, t_c) dt_c}{\nu^{(t)}(s)} & \text{if } \nu^{(t)}(s) \neq 0 \\ 0 & \text{otherwise} \end{cases}$$

---

**Algorithm 2** Online MCOT-C

**Input:** $\nu, N, (f_m)_{1 \leq m \leq M}, \kappa$
**Output:** V= {} the list of vehicles with their plugging time
S← {}
$\zeta^0 \leftarrow \mathbf{0_M}$
**for** $t$ **from** $0$ **to** $T$ **do**
   Add to S, vehicles that arrived at time $t$
   Compute $N_t$
   Update $\nu, \mu_1$ and $\mu_2$ as in Equations (13)
   $\zeta_m \leftarrow Alg(\zeta, \mu_1, \mu_2, y)$
   **for** $S_i$ in S **do**
      $t_c$ is generated according to $\mathrm{Mu}(\zeta, \mu_1, \mu_2, (S_i, .))$
      **if** $t_c = t$ **then**
         $f \leftarrow f - \frac{1}{N} f(S_i)$
         $S_i$ is removed from S and $(S_i, t_c)$ is added to V
      **end if**
   **end for**
**end for**

---

**Proposition 3.** **(i)** $\mathcal{E}_{t+1}(\zeta_t)$ *is bounded by $\kappa$, a stochastic term, and a term corresponding to a poor prediction of the law $\nu$:*

$$\mathcal{E}_{t+1}(\zeta_t) \leq \kappa + \left\| \left( \sum_{t_a^i = t+1} \frac{F_\zeta(S_i^{(t+1)})}{N} - \mathbb{E}_{\nu_r}[F_\zeta \mathbf{1}_{t_a = t+1}] \right)^+ \right\| + \left\| \left( \mathbb{E}_{\nu_r}[F_\zeta \mathbf{1}_{t_a = t+1}] - \mathbb{E}_{\nu}[F_\zeta \mathbf{1}_{t_a = t+1}] \right)^+ \right\|$$

**(i)** *The second term could be bounded with Bienaymé-Tchebychev inequality to obtain:*

$$\mathbb{P}\left( \left\| \left( \sum_{t_a^i = t+1} \frac{F_\zeta(S_i^{(t+1)})}{N} - \mathbb{E}_{\nu_r}[F_\zeta \mathbf{1}_{t_a = t+1}] \right)^+ \right\| \geq \kappa_0 \right) \leq \frac{\mathbb{V}_{\nu_r}[F_\zeta \mathbf{1}_{t_a = t+1}]}{N\kappa_0^2}$$

Thus, starting from scratch at each time step is unnecessary, and the optimization made in the previous step offers a good $\zeta$ to start with. This starting point is better if (i) the estimation of the arrival law of the vehicles $\nu$ is close from the real arrival law of vehicles $\nu_r$ and (ii) if $N$, the order of magnitude of EVs is large.

## 4.3 DATA OVERVIEW

The dataset used in this paper is composed of 10.000 random transactions from public charging stations operated by EVnetNL in the Netherlands (OpenDataset, 2019), in the year 2019. For each

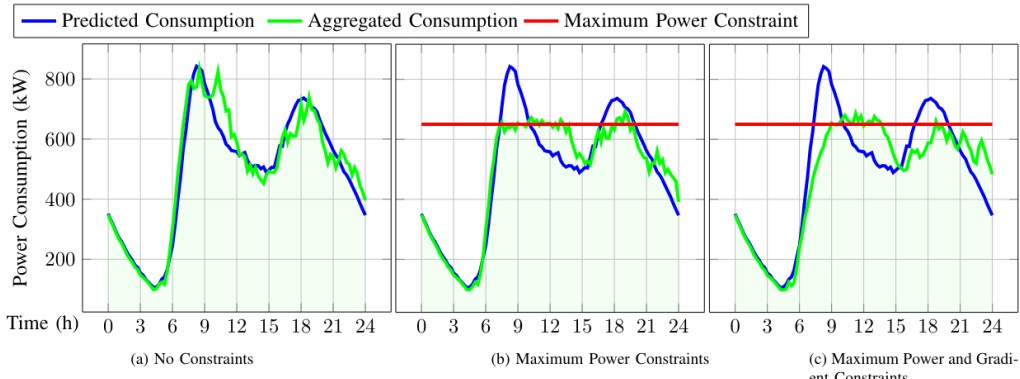

Figure 4: (a) Consumptions for the "Plug When Arrive" μ1 strategy with the arrival of EV predicted with ν and with the real distribution of EV; (b) Optimized Consumption for a constraint of 650kW for the aggregated consumption; (c) Optimized consumption for the same maximum power constraint and a constraint of 120kW/h for the gradient of the aggregated consumption.

transaction, several pieces of information are provided including the arrival time $t_a$, the leaving time $t_l$, the plugging time $\Delta t_n$, and the max power $P$. A more detailed description could be found in Refa & Hubbers (2019) and this dataset have already been used for clustering algorithm (Straka & Buzna, 2019) but not yet for Mean Field Control Algorithm.

There is a difference between weekdays and weekend days, so in this paper, we will consider the 7253 transactions happening during weekdays and divide them randomly. $90\%$ of these weekdays will form a training set of 231 days (6540 transactions) and will be considered historical data. A test day is created with the remaining $10\%$ of weekdays (21 days : 713 transactions) by grouping the corresponding 713 vehicle arrivals. The predicted distribution $\nu$ is computed on the training set considered historical data and $N = \frac{6540}{9} \simeq 727$ is the number of vehicles expected to arrive on this test day. In (6), we set $\varepsilon = 0.1$ because we want a relatively low value to limit the impact of entropic relaxation (term in Kullback Leibler), but not too low, as this risks posing computational problems (because of the $\varepsilon^{-1}$ in the exponential in Proposition 1.

To compute efficiently the gradient $G(\zeta_k)$ at each iteration of Algorithm 2, we need to discretize the state space $\mathcal{X}$: The day is divided into $T + 1 = 97$ steps (indexed from 0 to $T$) with a stepsize $\Delta t$ of 15 minutes, which allows rapid grid constraint changes to be taken into account. For the power discretization, we group each EV between 4kW, 7.5kW, and 12kW. This choice of discretization is standard (used for example in Sadeghianpourhamami et al. (2018)). We assume here that vehicles connected the day before are not affected by our strategy, because they are already connected, but their consumption is taken into account in order to come closer to reality, particularly in the case of controlling the gradient of aggregate consumption. We therefore consider the aggregate consumption of vehicles arriving throughout the day and that of vehicles arriving the day before (this impact is mainly present before 8 a.m.).

### 4.4 CONTROL OF THE AGGREGATED CONSUMPTION

On Fig. 4, the nominal consumption in blue corresponds to what is expected by the charging station, these are the historical data with the plugging strategy $\mu_1$ "Plug when Arrive". On (a), we can see the difference with the consumption for the real arrival of EV during the day with the same plugging strategy. The first peak in the morning lasts longer, while the second peak seems to be weaker. On (b), a constraint imposed by the charging station over the power consumed of $r_f = 650$kW is added through the moment constraints: define for each $m$ the function $f_m$ via $f_m(s, t_c) = p_{\max}$ if $m \in [t_c, t_c + \Delta t_n]$, $f_m(s, t_c) = 0$ otherwise, and impose for each $m$ the constraint $\langle f_m, \mu \rangle - r_f \leq 0$.

This value of 650kW is chosen arbitrarily here, and any other can be chosen as long as it remains realistic. This optimization makes it possible to exploit flexibility while respecting the imposed constraint, despite the prediction error on the length of the first peak. Peaks above the maximum constraint correspond to unforeseen arrivals of a large number of vehicles that must connect directly. It can also be due to the convergence not completely achieved by the algorithm, which depends on the value of $\kappa$ here chosen at 10kW.

## 4.5 CONTROL OF THE GRADIENT OF THE AGGREGATED CONSUMPTION

Another constraint that we want to respect in order to preserve the grid stability is the speed with which consumption will increase or decrease. On Fig. 4 (a) (b), we see a strong peak at the start of the day. We will seek to smooth this peak by imposing a constraint on the gradient of the power consumed. On (c), this constraint imposed by the charging station of $r_g = 100$kW/h is added through the moment constraints: $\forall m \in [0, T-1], \forall (s, t_c) \in \mathcal{X}^{(t)}, g_m(s, t_c) = f_{m+1}(s, t_c) - f_m(s, t_c)$ and we impose: $\forall m \in [0, T-1], -r_g \leq N \langle g_m, \mu \rangle \leq r_g$.

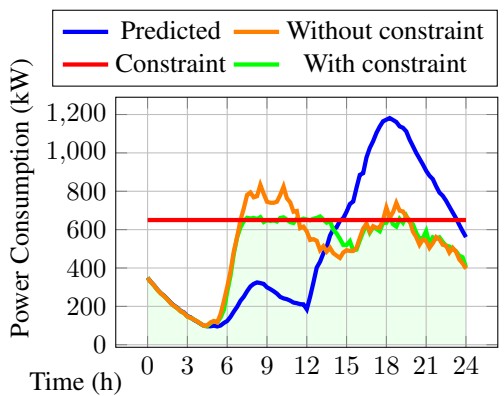

Figure 5: When the prediction $\nu$ differs greatly from the reality

This addition of constraints makes it possible to smooth out the slope which begins around 6am. There are always irregularities due to deviation from prediction and the slight excess of the constraint on the first peak can be explained by the maximum exploitation of the flexibility of the vehicles to respect the gradient constraint, which does not leave enough flexibility when vehicles arrive between 9am and 3pm and have to be connected directly.

## 4.6 SENSITIVITY TO THE DIFFERENCE BETWEEN ACTUAL EV ARRIVAL AND ITS PREDICTION

This model depends on the quality of the prediction $\nu$ made for the rest of the day. In this part, we try to test the robustness against this quality of prediction, by twisting the previous prediction: the central planner expects $30\%$ less vehicles before 12am and $30\%$ more vehicles after. The aggregated power consumption associated to this prediction is shown in blue in Fig. 5. We can thus observe that compliance with the same maximal power constraint of $650kW$ is still obtained and the consumption is very close to Fig. 4 (b). We therefore have a certain robustness of the model concerning the prediction $\nu$. This robustness is surely obtained here by the fact that we can change the connection time of a previously arrived vehicle as long as it is not connected. The algorithm can therefore, in the event of an unexpected arrival of vehicles to be connected immediately, postpone the connection time of less priority vehicles. But this poorer prediction comes at a cost: when comparing $\langle \pi, c \rangle$ between the case where the prediction is close (shown in figure 4 (a)) and this case, we find that the average time between arrival time $t_a$ and connection time $t_c$ increases from 11 minutes to 12 minutes. Having a less accurate prediction will therefore make less optimal use of flexibility.

## 5 CONCLUSIONS

One-sided moment relaxation of OT problem provides a very natural representation setting for tracking applications in control. In such applications, the OT problem is often infinite-dimensional (e.g. trajectories of agents). Instead of using approximations techniques for OT, MCOT-C leads to a tractable algorithm by directly considering only the distribution moments that are relevant for control. Furthermore, KL-term has a dual role in MCOT-C: a relaxation term as in many other machine learning algorithms, but it also enables to enforce the constraints on the dynamics via the choice of $\mu_2$ and absolute continuity imposed by KL. There are many directions for future research:

• The "Semi Sinkhorn" algorithm might be improved through the introduction of advanced optimization techniques (e.g., proximal methods or momentum).

• Obtain probabilistic error bounds for the stochastic gradient descent algorithms proposed in appendix E, which is useful in cases where the size of the problem makes the use of Monte Carlo methods attractive such as the water heaters problem presented in appendix E.2.

• We believe that representing distributions by their moments to perform optimal transport has broader applications in machine learning and control. We aim to explore its potential in other contexts.

**Reproducibility Statement** To ensure the reproducibility of scientific results, the code and the data used to obtain the results presented in this article are provided in the supplementary material. The theoretical proofs of the article as well as those given in the appendix A are presented in the appendix B.

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

In this appendix, dualization and proofs are presented in Section A and B. A theoretical extension is presented in appendix C, in the case where the distributions are Gaussian and the moments specified are the means and variances. In appendix D, an experiment involving the transport of a uniform law illustrates the convergence of the regularized problem to the non-regularized problem, when the regularization parameter $\varepsilon$ tends to 0. Another example of Mean Field Control using a Monte Carlo implementation is proposed in appendix E to illustrate the approach in the case of a large state space.

## A  DUALITY

First, we want to introduce 2 preliminary problems to the MCOT-C problem. The first problem is a variant of the relaxation of Alfonsi et al. (2020):

**Problem 1S-MCOT:** *One Sided Moment Constrained Optimal Transport.*

$$d(\mu_1, \mathcal{P}_f) = \min\{\langle \pi, c \rangle : \pi \in \mathcal{U}(\mu_1, \mu), \ \mu \in \mathcal{P}_f\} \tag{14}$$

Problem 1S-RMCOT is regularized using Kullback Leibler divergence:

**Problem 1S-RMCOT:** *One Sided - Regularized Moment Constrained Optimal Transport (1S-RMCOT).*

$$d_\varepsilon(\mu_1, \mathcal{P}_f) = \min_{\mu, \pi}\{\langle \pi, c \rangle + \varepsilon D_{\mathrm{KL}}(\pi \| \mu_1 \otimes \mu_2) : \pi \in \mathcal{U}(\mu_1, \mu), \ \mu \in \mathcal{P}_f\} \tag{15}$$

where $\varepsilon > 0$.

### A.1  DUAL FOR 1S-MCOT

Characterization of a solution to Problem 1S-MCOT is based on a Lagrangian relaxation. Introduce two classes of Lagrange multipliers for (14): $\psi$ is for the first marginal constraint, a real-valued measurable function on $\mathcal{X}$, and $\lambda \in \mathbb{R}_+^M$ for the moment constraints. The dual functional is defined as the infimum,

$$\varphi^*(\psi, \lambda) := \inf_\pi \ \langle \pi, c \rangle - \langle \pi_1 - \mu_1, \psi \rangle - \langle \pi_2, \lambda^\mathsf{T} h \rangle = \langle \mu_1, \psi \rangle + \inf_{x,y}\{c(x,y) - \psi(x) - \lambda^\mathsf{T} f(y)\} \tag{16}$$

The convex dual of (14) is defined to be the supremum of $\varphi^*(\psi, \lambda)$ over all $\psi$ and $\lambda$. The dual optimization problem admits a familiar representation. Compactness is assumed in Proposition 4 (ii), as in prior work concerning canonical distributions (Kemperman, 1968).

**Proposition 4.** *If (A1) and (A2) hold, then,*

**(i)** *With $\varphi^*$ defined in (16), the dual convex program admits the representation*

$$d^* := \sup_{\psi, \lambda} \varphi^*(\psi, \lambda) = \sup_{\psi, \lambda}\{\langle \mu_1, \psi \rangle : \psi(x) + \lambda^\mathsf{T} f(y) \leq c(x,y) \ \text{for all } x, y\} \tag{17}$$

*On replacing $\psi$ with $\psi^\lambda(x) := \inf_y\{c(x,y) - \lambda^\mathsf{T} f(y)\}$ we obtain the equivalent max-min problem*

$$d^* = \sup_\lambda \int \inf_y [c(x,y) - \lambda^\mathsf{T} f(y)] \mu_1(dx) \tag{18}$$

**(ii)** *Suppose in addition the set $\mathcal{X}$ is compact. Then the supremum in (17) is achieved, and there is no duality gap: for a vector $\lambda^* \in \mathbb{R}_+^M$,*

$$d(\mu_1, \mathcal{P}_f) = d^* = \int \min_y\{c(x,y) - \lambda^{*\mathsf{T}} f(y)\} \mu_1(dx)$$

We present here the proof of part (i). The proof of (ii) is based on approximation with solutions to 1S-RMCOT. A summary of the approach is contained in Proposition 6.  □

Once we solve (17), we obtain $\pi^*$ through complementary slackness:

$$0 = \sum_{x,y} \pi^*(x,y)\{\psi^*(x) + \lambda^{*\mathsf{T}} f(y) - c(x,y)\}$$

which means that $\pi^*$ is supported on the set $\{(x,y) : \lambda^{*\mathsf{T}} f(y) + \psi^*(x) = c(x,y)\}$.

## A.2 REGULARIZATION

Recall that the functional $D_{\mathrm{KL}}(\pi\|\mu_1 \otimes \mu_2)$ is used to define the Sinkhorn distance (Cuturi, 2013), and coincides with mutual information when the marginals of $\pi$ agree with the given probability measures $\mu_1$ and $\mu_2$. In the present paper, the marginal $\mu_2$ is a design parameter.

**1S-RMCOT geometry and duality** A close cousin to 1S-RMCOT uses the Kullback Leibler divergence as a constraint rather than penalty (Cuturi, 2013). Consider for fixed $\delta > 0$,

$$d_\delta^c(\mu_1, \mathcal{P}_f) = \min\ \langle \pi, c \rangle, \quad \text{s.t. } \pi \in \mathcal{U}(\mu_1, \mu), \mu \in \mathcal{P}_f,\ D_{\mathrm{KL}}(\pi\|\mu_1 \otimes \mu_2) \leq \delta \qquad (19)$$

The parameter $\varepsilon > 0$ in (15) may be regarded as a Lagrange multiplier corresponding to the constraint $D_{\mathrm{KL}}(\pi\|\mu_1 \otimes \mu_2) \leq \delta$. Under general conditions there is $\delta(\varepsilon)$ such that the optimizers of (19) and (15) coincide.

In considering the dual of (15) we choose a relaxation of the moment constraints only: letting $\lambda \in \mathbb{R}_+^M$ denote the Lagrange multiplier as before,

$$\varphi^*(\lambda) := \inf_\pi \{\langle \pi, c \rangle + \varepsilon D_{\mathrm{KL}}(\pi\|\mu_1 \otimes \mu_2) - \langle \pi_2, \lambda^\mathsf{T} h \rangle : \pi_1 = \mu_1\} \qquad (20)$$

The convex dual of 1S-RMCOT is by definition the supremum of the concave function $\varphi^*$. The optimizer, when it exists, is denoted $\pi^\lambda$.

Introducing the notation

$$\ell_0^\lambda(x, y) = \lambda^\mathsf{T} f(y) - c(x, y), \quad x, y \in \mathcal{X} \qquad (21)$$

the dual function may be expressed

$$\varphi^*(\lambda) = -\max_\pi \{\langle \pi, \ell_0^\lambda \rangle - \varepsilon D_{\mathrm{KL}}(\pi\|\mu_1 \otimes \mu_2) : \pi_1 = \mu_1\}$$

The dual of (19) with $d = d(\varepsilon)$ yields better geometric insight. If the maximum above exists, then the maximizer $\pi^\lambda$ solves

$$\pi^\lambda \in \arg\max\{\langle \pi, \ell_0^\lambda \rangle : D_{\mathrm{KL}}(\pi\|\mu_1 \otimes \mu_2) \leq \delta,\ \pi_1 = \mu_1\}$$

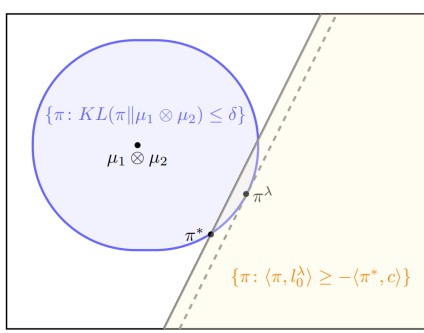

Figure 6: Dual geometry for OT-FPR

The convex region containing $\mu_1 \otimes \mu_2$ shown in Fig. 6 is the set of all $\pi$ for which $\pi_1 = \mu_1$ and $D_{\mathrm{KL}}(\pi\|\mu_1 \otimes \mu_2) \leq \delta$. The optimizer $\pi^\lambda$ lies on the intersection of this region and the hyperplane shown in the figure, indicated with a dashed line: $\{\pi : \langle \pi, \ell_0^\lambda \rangle = \langle \pi^\lambda, \ell_0^\lambda \rangle\}$. This value of $\lambda$ does not optimize $\varphi^*$ because the hyperplane is not the boundary of the half-space shown in the figure.

For computation, it is convenient to make a change of variables: since $\pi_1 = \mu_1$ is constrained, the infimum is over all probability kernels: for $\lambda \in \mathbb{R}_+^M$,

$$\varphi^*(\lambda) := \inf_T \{-\langle \mu_1 T, \ell_0^\lambda \rangle + \varepsilon D_{\mathrm{KL}}(\mu_1 T\|\mu_1 \otimes \mu_2)\} \qquad (22)$$

For each $\lambda \in \mathbb{R}_+^M$, $\varepsilon > 0$ and $x \in \mathcal{X}$, denote

$$B_{\lambda,\varepsilon}(x) = \varepsilon \log \int_{y \in \mathcal{X}} \exp\left(\varepsilon^{-1} \ell_0^\lambda(x, y)\right) \mu_2(dy) \qquad (23)$$

**Proposition 5.** *Subject to (A1)–(A3),*

**(i)** *The infimum (22) gives* $\varphi^*(\lambda) = -\langle \mu_1, B_{\lambda,\varepsilon} \rangle$.

**(ii)** *The probability kernel maximizing (22) is*

$$T^\lambda(x, dy) = \mu_2(dy) \exp(L^\lambda(x, y)), \text{with } L^\lambda(x, y) = \varepsilon^{-1}\{\ell_0^\lambda(x, y) - B_{\lambda,\varepsilon}(x)\} \qquad (24a)$$

**(iii)** *unique $\lambda^* \in \mathbb{R}_+^M$ exists, satisfying*

$$\varphi^*(\lambda^*) = d_\varepsilon(\mu_1, \mathcal{P}_f) \tag{24b}$$

*That is, there is no duality gap.*

The similarity between Proposition 5 and Proposition 4 is found through examination of (18), and the recognition that $-B_{\lambda,\varepsilon}(x)$ is a ($\mu_2$-weighted) soft minimum of $-\ell_0^\lambda(x,y) = c(x,y) - \lambda^\mathsf{T} f(y)$ over $y \in \mathcal{X}$. Subject to this interpretation, the convex dual of 1S-RMCOT can be expressed in a form entirely analogous to (18):

$$\max_\lambda \varphi^*(\lambda) = \max_\lambda \int \operatorname*{softmin}_y \{c(x,y) - \lambda^\mathsf{T} f(y)\} \mu_1(dx)$$

**1S-MCOT approximation**

Consider the following procedure to obtain a solution to 1S-MCOT (without regularization), but with $\mathcal{X}$ compact, and the supports of $\mu_1$ and $\mu_2$ each equal to all of $\mathcal{X}$. Let $\{\pi^\varepsilon, \lambda^\varepsilon : \varepsilon > 0\}$ denote primal-dual solutions to 1S-RMCOT, where $\varepsilon > 0$ is the scaling in (15). Hence for each $\varepsilon > 0$,

$$d_\varepsilon(\mu_1, \mathcal{P}_f) = \langle \pi^\varepsilon, c \rangle + \varepsilon D_{\mathrm{KL}}(\pi^\varepsilon \| \mu_1 \otimes \mu_2) = -\langle \mu_1, B_{\lambda^\varepsilon, \varepsilon} \rangle$$

**Proposition 6.** *Suppose that the assumptions of Proposition 4 (ii) hold, so in particular $\mathcal{X}$ is compact. Then, any weak subsequential limit of $\{\pi^\varepsilon, \lambda^\varepsilon : \varepsilon > 0\}$ as $\varepsilon \downarrow 0$ defines a pair $(\pi^0, \lambda^0)$ for which $\pi^0$ solves 1S-MCOT and $\lambda^0$ achieves the supremum in (18).*

*Furthermore, it is possible to bound the rate of convergence:*

$$|d_\varepsilon^*(\mu_1, \mathcal{P}_f) - d^*(\mu_1, \mathcal{P}_f)| \leq \varepsilon D_{\mathrm{KL}}(\pi^0 \| \mu_1 \otimes \mu_2)$$

## A.3 LINK WITH THE MCOT-C PROBLEM

Writing the dual of MCOT-C, we get:

$$\varphi^*(\lambda) = \varepsilon \min_{\pi, \mu} \{ -\langle \pi, l \rangle + D_{\mathrm{KL}}(\pi \| \mu_1 \otimes \mu_2) : \pi \in K(\mu_1, \mu) \}$$

Since $\pi \in K(\mu_1, \mu)$ is constrained, the infimum is over all probability kernels $T$ from $\mathcal{X}$ to $\mathcal{W}$:

$$\varphi^*(\lambda) = -\int_x \mu_1(dx) \max_{T(x,.)} \{ \langle T(x,.), \ell_0^\lambda(x,.) \rangle_\mathcal{W} - \varepsilon D_{\mathrm{KL}}(T(x,.) \| \mu_2(s^x,.)) \}$$

where $\langle .,. \rangle_\mathcal{W}$ is the inner product on $\mathcal{W}$. We obtain Proposition 1, which gives similar results as Prop. 5 with a probability kernel going from $\mathcal{X}$ to $\mathcal{W}$.

## B PROOFS

Much of the analysis that follows is based on convex duality between relative entropy and log moment generating functions. For any probability measure $\mu$ on $\mathcal{X}$ and function $g \colon \mathcal{X} \to \mathbb{R}$, the log moment generating function is denoted,

$$\Lambda_\mu(g) = \log \langle \mu, e^g \rangle$$

With $\mu$ fixed, this is viewed as an extended-valued, convex functional on the space of Borel measurable functions. Lemma 2 is a standard tool in information theory (Dembo & Zeitouni, 1998), and a reason that relative entropy is popular for use as a regularizer in optimization.

**Lemma 2.** *Relative entropy and the log moment generating function are related via convex duality:*

*For any probability measure $p$ we have*

$$D_{KL}(p \| \mu) = \sup_g \{ \langle p, g \rangle - \Lambda_\mu(g) \} \tag{25a}$$

*If $D_{KL}(p \| \mu) < \infty$ then the supremum is achieved, with optimizer equal to the log likelihood ratio, $g^* = \log(dp/d\mu)$.*

*For Borel measurable $g \colon \mathcal{X} \to \mathbb{R}$,*

$$\Lambda_\mu(g) = \sup_p \{\langle p, g\rangle - D_{KL}(p\|\mu)\} \tag{25b}$$

*If $\Lambda_\mu(g) < \infty$ then the supremum is achieved, where the optimizer $p^*$ has log likelihood ratio $\log(dp^*/d\mu) = g - \Lambda_\mu(g)$.* □

**Proof of Proposition 5**  For each $\lambda$ we have by definition,

$$\varphi^*(\lambda) = \min_T \int_{x\in\mathcal{X}} \mu_1(dx)\Big\{\varepsilon D_{\mathrm{KL}}(T(x,\cdot)\|\mu_2) - \int_{y\in\mathcal{X}} T(x,dy)\ell_0^\lambda(x,y)\Big\} \tag{26}$$

$$= -\varepsilon \max_T \int_{x\in\mathcal{X}} \mu_1(dx)\Big\{\varepsilon^{-1} \int_{y\in\mathcal{X}} T(x,dy)\ell_0^\lambda(x,y) - D_{\mathrm{KL}}(T(x,\cdot)\|\mu_2)\Big\} \tag{27}$$

For each $x$ we have an optimization problem of the form (25b). Applying Lemma 2 (ii) gives the representation (9a) and by substitution (or applying (25b)) we obtain

$$\Big\{\varepsilon^{-1} \int_{y\in\mathcal{X}} T^\lambda(x,dy)\ell_0^\lambda(x,y) - D_{\mathrm{KL}}(T^\lambda(x,\cdot)\|\mu_2)\Big\} = \varepsilon^{-1}B_{\lambda,\varepsilon}(x) \tag{28}$$

Integrating with respect to $\mu_1$ and applying (27) completes the proof. □

**Proof of Proposition 1**  The proof is the same as the previous one using this expression of the dual:

$$\varphi^*(\lambda) = -\int_x \mu_1(dx) \max_{T(x,.)}\big\{\langle T(x,.), \ell_0^\lambda(x,.)\rangle_\mathcal{W} - \varepsilon D_{\mathrm{KL}}(T(x,.)\|\mu_2(s^x,.))\big\}$$

**Proof of Proposition 6**  Let $(\pi^\varepsilon, \lambda^\varepsilon)$ denote the solution to 1s-RMCOT, with $\varepsilon > 0$ regarded as a variable. We let $(\pi^0, \lambda^0)$ denote any weak sub-sequential limit: for a sequence $\{\varepsilon_i \downarrow 0\}$,

$$\pi^{\varepsilon_i} \to \pi^0, \qquad \lambda^{\varepsilon_i} \to \lambda^0, \qquad i \to \infty.$$

Optimality of $\pi^0$ is established in the following steps:

- Subject to (A1) and (A2) we know that $\pi^0 \in \mathcal{U}(\mu_1, \mu)$ with $\mu \in \mathcal{P}_f$.

- For any $\pi \in \mathcal{U}(\mu_1, \mu)$ with $\mu \in \mathcal{P}_f$ and $D_{\mathrm{KL}}(\pi\|\mu_1 \otimes \mu_2) < \infty$ and any $\varepsilon > 0$ we have

$$\langle \pi^0, c\rangle = \lim_{i\to\infty} \langle \pi^{\varepsilon_i}, c\rangle \leq \lim_{i\to\infty}\{\langle \pi^{\varepsilon_i}, c\rangle + \varepsilon_i D_{\mathrm{KL}}(\pi^{\varepsilon_i}\|\mu_1\otimes\mu_2)\} \leq \lim_{i\to\infty}\{\langle \pi, c\rangle + \varepsilon_i D_{\mathrm{KL}}(\pi\|\mu_1\otimes\mu_2)\} = \langle \pi, c\rangle$$

- Under the support assumption we can approximate in the weak topology any $\pi \in \mathcal{U}(\mu_1, \mu)$ with $\mu \in \mathcal{P}_f$ by $\pi^\delta$ satisfying $D_{\mathrm{KL}}(\pi^\delta\|\mu_1 \otimes \mu_2) < \infty$ and

$$\langle \pi^0, c\rangle \leq \langle \pi^\delta, c\rangle \leq \langle \pi, c\rangle - \delta$$

Since $\delta > 0$ is arbitrary this establishes optimality.

We next show $\lambda^0$ provides an optimal solution. Proposition 3.2 gives for any $\lambda$,

$$\langle \pi^0, c\rangle \geq -\lim_{i\to\infty}\langle \mu_1, B_{\lambda,\varepsilon_i}\rangle = \int \in f_y\{c(x,y) - \lambda^T f(y)\}\mu_1(dx)$$

The lower bound is achieved using $\lambda^0$ by allowing $\lambda$ to depend on $i$:

$$\langle \pi^0, c\rangle \leq \lim_{i\to\infty}\{\langle \pi^{\varepsilon_i}, c\rangle + \varepsilon_i D_{\mathrm{KL}}(\pi^{\varepsilon_i}\|\mu_1\otimes\mu_2)\} = -\lim_{i\to\infty}\langle \mu_1, B_{\lambda^{\varepsilon_i},\varepsilon_i}\rangle = \int \inf_y\{c(x,y) - \lambda^{0^T} f(y)\}\mu_1(dx)$$

To prove the rate of convergence, we adapt results from Luise et al. (2018) in our context. First, we denote $\pi_\varepsilon = argmin[\langle \pi, c\rangle + \varepsilon D_{\mathrm{KL}}(\pi\|\mu_1 \otimes \mu_2)]$ and by optimality of $\pi_\varepsilon$, we obtain: $\langle \pi_\varepsilon, c\rangle + \varepsilon D_{\mathrm{KL}}(\pi_\varepsilon\|\mu_1 \otimes \mu_2) \leq \langle \pi_0, c\rangle + \varepsilon D_{\mathrm{KL}}(\pi_0\|\mu_1 \otimes \mu_2)$

By optimality of $\pi_0$ and positivity of the Kullback Leibler divergence, we obtain: $\langle \pi_0, c \rangle \leq \langle \pi_\varepsilon, c \rangle \leq \langle \pi_\varepsilon, c \rangle + \varepsilon D_{\mathrm{KL}}(\pi_\varepsilon \| \mu_1 \otimes \mu_2)$

Combining these inequalities, we get:

$$0 \leq \langle \pi_\varepsilon, c \rangle + \varepsilon D_{\mathrm{KL}}(\pi_\varepsilon \| \mu_1 \otimes \mu_2) - \langle \pi_0, c \rangle \leq \varepsilon D_{\mathrm{KL}}(\pi_0 \| \mu_1 \otimes \mu_2)$$

$$0 \leq d_\varepsilon^*(\mu_1, \mathcal{P}_f) - d^*(\mu_1, \mathcal{P}_f) \leq \varepsilon D_{\mathrm{KL}}(\pi_0 \| \mu_1 \otimes \mu_2)$$

which proves our result.

**Proof of Lemma 1**  Suppose that $v \in \mathbb{R}^M$ is in the null space: $\Sigma^\lambda v = 0$. From the definition (10b) it follows that

$$0 = v^\mathsf{T} \Sigma^\lambda v = \mathsf{E}^\lambda \big[ \big\{ v^\mathsf{T} \big( f(Y) - \mathsf{E}^\lambda[f(Y) \mid X] \big) \big\}^2 \big]$$

Equivalently, there is a function $g \colon \mathcal{X} \to \mathbb{R}$ such that

$$v^\mathsf{T} f(Y) = g(X) \qquad a.s. \ [\pi^\lambda]$$

The probability measures $\pi^\lambda$ and $\pi^0 := \mu_1 \otimes \mu_2$ are mutually absolutely continuous, so the same equation holds under a.s. $[\pi^0]$. Independence gives

$$v^\mathsf{T} f(Y) = \mathsf{E}^0[v^\mathsf{T} f(Y) \mid Y] = \mathsf{E}^0[g(X) \mid Y] = \langle \mu_1, g \rangle \qquad a.s. \ [\pi^0]$$

That is, the variance of $v^\mathsf{T} f(Y)$ is equal to zero. Under (A3) this is possible only if $v = 0$. $\qquad\square$

**Proof of Proposition 2**  Recall the notation $\mu^\lambda = \mu_1 T^\lambda$, which is the second marginal of $\pi^\lambda$, and the probabilistic notation defined in the Introduction. Also, by definition we have $\mathcal{J}(\zeta) = \varepsilon^{-1} \langle \mu_1, B_{\varepsilon\zeta,\varepsilon} \rangle$.

We have for each $i$,

$$\varepsilon^{-1} \frac{\partial}{\partial \zeta_i} B_{\varepsilon\zeta,\varepsilon}(x) = \frac{\int_{y \in \mathcal{X}} \mu_2(y) \exp\big( \{ \zeta^\mathsf{T} f(y) - \varepsilon^{-1} c(x,y) \} \big) f^m(y)}{\int_{y \in \mathcal{X}} \mu_2(y) \exp\big( \{ \zeta^\mathsf{T} f(y) - \varepsilon^{-1} c(x,y) \} \big)} = T^\lambda f^m(x)$$

Integrating each side over $\mu_1$ gives (10a) (recall that $\mu^\lambda = \mu_1 T^\lambda$).

To obtain the second derivative of $\mathcal{J}(\zeta)$ requires the first derivative of the log-likelihood:

$$L_j^{\varepsilon\zeta}(x,y) := \frac{\partial}{\partial \zeta_j} L^{\varepsilon\zeta}(x,y) = \frac{\partial}{\partial \zeta_j} \big[ \zeta^\mathsf{T} f(y) - \varepsilon^{-1} B_{\varepsilon\zeta,\varepsilon}(x) \big] = h_j(y) - T^\lambda h_j(x)$$

From this we obtain,

$$
\begin{aligned}
\frac{\partial^2}{\partial \zeta_i \partial \zeta_j} B_{\varepsilon\zeta,\varepsilon}(x) &= \frac{\partial}{\partial \zeta_j} T^{\varepsilon\zeta} f^m(x) \\
&= \int T^{\varepsilon\zeta}(x, dy) \{ L_j^{\varepsilon\zeta}(x,y) f^m(y) \} \\
&= \int T^{\varepsilon\zeta}(x, dy) h_j(y) f^m(y) - T^\lambda h_j(x) \int T^{\varepsilon\zeta}(x, dy) h_j(y) \\
&= \mathsf{E}^\lambda[h_j(Y) f^m(Y) \mid X = x] - \mathsf{E}^\lambda[f^m(Y) \mid X = x] \mathsf{E}^\lambda[h_j(Y) \mid X = x]
\end{aligned}
$$

Integrating each side over $\mu_1$ gives (10b). $\qquad\square$

**Proposition 7.** *The conditional distribution defined in* (9a) *is Markovian: for a collection of probability kernels* $\{ \check{P}_i^\lambda \}$ *parameterized by* $x$,

$$T^\lambda(x, dy) = \nu_0(dy_0) \prod_{i=1}^{M} \check{P}_i^\lambda(y_{i-1}, dy_i; x) \tag{29}$$

**Proof of Proposition 7**  The proof reduces to justifying (29), which is one component of Proposition 8 that follows.

Write $L_i^\lambda(x_i, y_i) = \varepsilon^{-1}\{\lambda_i(\mathcal{U}(y_i) - r_i) - \frac{1}{2}\|x_i - y_i\|^2\}$, and for each $i$ consider the positive kernel,

$$\widehat{P}_i^\lambda(y_{i-1}, dy_i) = P_i(y_{i-1}, dy_i)\exp\big(L_i^\lambda(x_i, y_i)\big)$$

**Proposition 8.** *The conditional distribution defined in* (9a) *can be expressed*

$$T^\lambda(x, dy) = \nu_0(dy_0)\exp\big(-\varepsilon^{-1}B_{\lambda,\varepsilon}(x)\big)\prod_{i=1}^M \widehat{P}_i^\lambda(y_{i-1}, dy_i) \tag{30}$$

*Consequently, conditioned on $X = x$, the process $Y$ is of the form* (29), *in which each kernel in the product takes the form,*

$$\check{P}_i^\lambda(y_{i-1}, dy_i; x) = \frac{1}{g_{i-1}(y_{i-1}; x)}\widehat{P}_i^\lambda(y_{i-1}, dy_i)g_i(y_i; x)$$

*The functions $\{g_i : 0 \leq i \leq M\}$ are defined inductively: $g_M(y_M; x) \equiv 1$, and for $1 \leq i \leq M$,*

$$g_{i-1}(y; x) := \int \widehat{P}_i^\lambda(y, dy_i)g_i(y_i; x), \quad y \in \mathsf{X}$$

*This results in $g_0(y_0, x) = \exp\big(\varepsilon^{-1}B_{\lambda,\varepsilon}(x)\big)$.*

**Proof**  The representation (30) follows from the definition (9a) and the structure imposed on $h$ and $\mu_1$. It is then immediate that (30) can be transformed to (29): by construction,

$$\prod_{i=1}^M \check{P}_i^\lambda(y_{i-1}, dy_i; x) = \frac{1}{g_0(y_0; x)}\prod_{i=1}^M \widehat{P}_i^\lambda(y_{i-1}, dy_i)$$

Since $y_0 = x_0$ by construction, it also follows that

$$\exp\big(\varepsilon^{-1}B_{\lambda,\varepsilon}(x)\big) = g_0(x_0; x)$$

$$\square$$

## C  EXAMPLE: QUADRATIC CONSTRAINTS & GAUSSIAN REGULARIZER

Consider the special case in which the function $f$ is designed to specify all first and second moments for $Y$. To solve Problem 2 we adopt the following notational conventions for the Lagrange multiplier: $\mathsf{E}[Y_i] = m_i^1 \longleftrightarrow \lambda_i^1$ and $\mathsf{E}[Y_iY_j] = m_{ij}^2 \longleftrightarrow \lambda_{i,j}^2$. Of course we have $m_{ij}^2 = m_{ji}^2$ for each $i, j$. The total number of constraints is thus $M = n + n(n + 1)/2$. For purposes of calculation it is useful to introduce the symmetric matrices $M_Y^2$ and $\Lambda^2$ with respective entries $\{m_{ij}^2\}$ and $\{\lambda_{ij}^2\}$; similar notation is used for $m_Y$ and $\lambda^1$, the $n$-dimensional vectors with entries $\{m_i^1\}$ and $\{\lambda_i^1\}$.

Eq. (8) gives $\ell_0^\lambda(x, y) = \lambda^\mathsf{T}f(y) - c(x, y)$ with

$$\lambda^\mathsf{T}f(y) = y^\mathsf{T}\Lambda^2 y - \langle\Lambda^2, M_Y^2\rangle + y^\mathsf{T}\lambda^1 - m_Y^\mathsf{T}\lambda^1 \tag{31}$$

An explicit solution to problem 1S-RMCOT is obtained when $c$ is quadratic and $\mu_2$ is Gaussian:

**Proposition 9.** *Consider the 1S-RMCOT optimization problem* (15) *in the following special case: $c(x, y) = \frac{1}{2}\|x - y\|^2$, and $\mu_2 = N(0, I)$ in the regularizer* (5). *Assume that the target covariance $\Sigma_Y := M_Y^2 - m_Y m_Y^\mathsf{T}$ is positive definite.*

*Then, for each $\lambda$ with $\Lambda^2 < \frac{1}{2}(1 + \varepsilon)I$, the probability kernel $T^\lambda$ is Gaussian: conditioned on $X = x$, the distribution of $Y$ is Gaussian $N(m_{T^\lambda}^x, \Sigma_{T^\lambda})$ with*

$$m_{T^\lambda}^x = \varepsilon^{-1}\Sigma_{T^\lambda}[x + \lambda^1], \quad \Sigma_{T^\lambda} = \big[I + \varepsilon^{-1}[I - 2\Lambda^2]\big]^{-1} \tag{32}$$

**Proof of Proposition 9**  From (31) and using $c(x,y) = \frac{1}{2}\|x-y\|^2$ we obtain an expression for the likelihood $L^\lambda$ appearing in (9a):

$$L^\lambda(x,y) = \varepsilon^{-1}\left\{y^\mathsf{T}\Lambda^2 y + y^\mathsf{T}\lambda^1 - \kappa^\lambda - B_{\lambda,\varepsilon}(x)\right\} - \tfrac{1}{2}(\|x\|^2 - 2x^\mathsf{T}y + \|y\|^2)\} \tag{33}$$

with $\kappa^\lambda = \langle \Lambda^2, M_Y^2 \rangle + m_Y^\mathsf{T}\lambda^1$. The expression for $T^\lambda$ in (9a) using $\mu_2 = N(0,I)$ then implies that for any $x$, $T^\lambda(x, dy)$ admits the Gaussian density

$$\tau^\lambda(y \mid x) = \frac{1}{n^\lambda(x)}\exp\!\left(-\tfrac{1}{2}\|y\|^2\right)\exp\!\left(\varepsilon^{-1}\{-\tfrac{1}{2}y^\mathsf{T}[I - 2\Lambda^2]y + y^\mathsf{T}[x + \lambda^1]\}\right) \tag{34}$$

where $n^\lambda(x) = (2\pi)^{n/2}\exp\!\left(\varepsilon^{-1}\{\kappa^\lambda + B_{\lambda,\varepsilon}(x) + \tfrac{1}{2}\|x\|^2\}\right)$ may be regarded as a normalizing constant. $\qquad\square$

**Computation for non-Gaussian $\mu_1$**  In this case it is necessary to compute the normalizing constant in the definition of $T^\lambda$:

$$n^\lambda(x) = n^\lambda(x)\int \tau^\lambda(y \mid x)\,dy = \int \exp\!\left(-\tfrac{1}{2}y^\mathsf{T}\Sigma_{T^\lambda}^{-1}y + \varepsilon^{-1}y^\mathsf{T}[x+\lambda^1]\right)dy \tag{35}$$

$$= \sqrt{(2\pi)^d \det(\Sigma_{T^\lambda})}\exp\!\left(\tfrac{1}{2}\varepsilon^{-2}[x+\lambda^1]^\mathsf{T}\Sigma_{T^\lambda}[x+\lambda^1]\right) \tag{36}$$

Monte-Carlo methods can be used to estimate $\lambda^*$. Denote for each $x$,

$$q^\lambda(x) = \int T^\lambda(x, dy)f(y)\,, \quad m^\lambda(x) = \int T^\lambda(x, dy)f(y)f(y)^\mathsf{T}$$

Each have polynomial entries: $q_i^\lambda$ is a quadratic function of $x$ and $m_{i,j}^\lambda(x)$ is a fourth order polynomial in $x$ for each $i, j$. In applying any of the algorithms described in Section E one might take

$$\widetilde{m}^{n+1} = q^{\lambda_n}(X_{n+1})\,, \qquad \widetilde{\Sigma}^{n+1} = m^{\lambda_n}(X_{n+1}) - \widetilde{m}^{n+1}[\widetilde{m}^{n+1}]^\mathsf{T}$$

These functions will have finite means provided $\mathsf{E}[\|X\|^4]$ is finite under $\mu_1$.

# D  CONVERGENCE RATE WHEN TRANSPORTING FROM A UNIFORM DISTRIBUTION

We want to illustrate the convergence rate in Proposition 6.

With the same notations as in problems 1S-MCOT and 1S-RMCOT, we define $\mathcal{X} = [0, 1]$. Distributions $\mu_1$ and $\mu_2$ are the uniform distribution on $\mathcal{X}$. We define $f(x) = x - m$ with $m \in \mathcal{X}$ the imposed mean, and impose a unique constraint: $\langle f, \mu \rangle = 0$. The cost $c$ is chosen as : $\forall x, y \in \mathcal{X}, c(x, y) = (x - y)^2$.

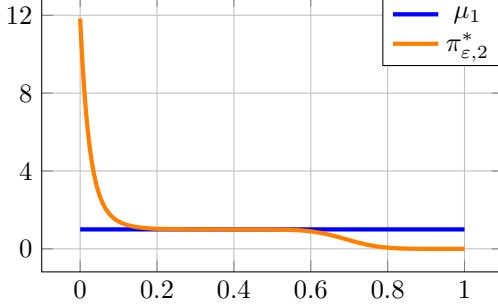

Figure 7: For $\varepsilon = 0.01$, $\mu_1$ is transported to $\pi_2$ with mean $0.25$

For these values, it is possible to obtain an explicit solution to 1S-MCOT, using Proposition 3.1:

$$d^* = \sup_\lambda \int \inf_y [c(x,y) - \lambda f(y)]\mu_1(dx) = \sup_\lambda \int \inf_y [(x-y)^2 - \lambda(y - m)]dx$$

$$= \sup_\lambda \int -\frac{\lambda^2}{4} + \lambda(m - x)dx$$

$$= (m - 0.5)^2$$

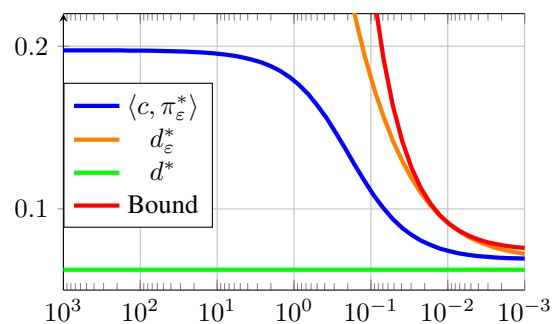

Figure 8: Comparison of the costs $d^*$ and $\langle c, \pi_\varepsilon^* \rangle$ for different values of $\varepsilon$

The solution $\pi_\varepsilon^*$ may be obtained through gradient descent as explained in section 3. For $m = 0.25$ and a discretization of $\mathcal{X}$ to 100 points (to compute the gradient), the resulting marginal $\pi_2$ is shown in Fig. 7, achieving the constraint on the mean.

The values of $d^*$ and $\langle c, \pi_\varepsilon^* \rangle$, were obtained for a range of $\varepsilon$ (from $10^{-3}$ to $10^3$). We can observe in Fig. 8 that the convergence to the minimum of the unregularized problem is fast and that it respects the inequality proved in Proposition 6:

$$|d_\varepsilon^*(\mu_1, \mathcal{P}_f) - d^*(\mu_1, \mathcal{P}_f)| \leq \varepsilon D_{\mathrm{KL}}(\pi_0 \| \mu_1 \otimes \mu_2)$$

# E    MONTE CARLO METHODS TO ACCELERATE THE CONVERGENCE OF A WATER HEATER CONTROL PROBLEM

## E.1    STOCHASTIC GRADIENT DESCENT

The gradient computation at each iteration of Algorithm 1 is of complexity $N_\mathcal{S} \times N_\mathcal{W}^2$, where $N_\mathcal{S}$ and $N_\mathcal{W}$ are respectively the number of discretization points for $\mathcal{S}$ and $\mathcal{W}$. Thus, when these numbers are large, it could be useful to use a stochastic gradient descent to approximate $\zeta^*$. Suppose that $\{X_n\}$ is i.i.d. $[\mu_1]$, and given an estimate $\zeta^k \in \mathbb{R}_+^M$ and the observation $x = X_{k+1}$, we draw $Y_{k+1} \sim T^{\varepsilon^{-1}\zeta_k}(x, \cdot)$ independently of $\{(X_l, Y_l) : l \leq k\}$. Given an initial condition $\zeta^0 \in \mathbb{R}_+^M$, a non-negative step-size sequence $\{\rho_n\}$, and positive definite matrices $\{G^n\}$, the projected stochastic gradient descent algorithm is defined by the recursion

$$\zeta^{k+1} = (\zeta^k - \rho_{k+1} G^{k+1} \widetilde{m}^{k+1})_+ \tag{37}$$

in which $\mathsf{E}[\widetilde{m}_i^{k+1} \mid \lambda_k] = \langle \pi_2^{\lambda^k}, f^m \rangle$ for each $i$ and $(.)_+$ is defined by: $\forall x \in \mathbb{R}^M, (x)_+ = \{\max(x_i, 0)\}_{i \in [1, M]}$.

If $M$ is not large we might opt for Zap stochastic approximation (Devraj et al., 2021), in which $G^n$ approximates the inverse of $\nabla^2 \mathcal{J}(\lambda^n)$. This can be achieved using $G^n = [\overline{\Sigma}^n]^{-1}$ for each $n$, where the estimates evolve as

$$\overline{\Sigma}^{k+1} = \overline{\Sigma}^k + \mu_{2,k+1}\{\widetilde{\Sigma}^{k+1} - \overline{\Sigma}^n\} \tag{38}$$

initialized with $\overline{\Sigma}^0 > 0$ (positive definite), and where $\widetilde{\Sigma}^{k+1}$ is random with conditional mean $\mathsf{E}[\widetilde{\Sigma}^{k+1} \mid \lambda_k] = \Sigma^{\lambda_k}$ (see (10b)). The stepsize sequence is chosen with $\mu_{2,n} \gg \rho_k$ for large $k$ (Devraj et al., 2021).

Two choices for the construction of $\{\widetilde{\Sigma}^{k+1}\}$ are summarized in the following:

**1. Conditional computation plus sampling:** In each of the general examples described in Section 3 it is not computationally expensive to compute the conditional means $m := T^\lambda f$ and second moments $m^2 := T^\lambda[f^\intercal f]$. In this case we draw $\{X_k\}$ i.i.d., and take for each $k$,

$$\widetilde{m}^{k+1} = m(X_{k+1}), \quad \widetilde{\Sigma}^{k+1} = m^2(X_{k+1}) - \widetilde{m}^{k+1}[\widetilde{m}^{k+1}]^\intercal$$

**2. Split Sampling:** This approach requires that we obtain $\{X_k\}$ i.i.d., and also be able to draw a sample from the probability kernel. With $Z \geq 2$ an integer: draw $x = X_{k+1}$ from $\mu_1$, and then draw $Z$ independent samples $\{Y_{k+1}^z : 1 \leq z \leq Z\}$ from $T^\lambda(x, \cdot)$, independently of $\{(X_z, Y_z) : z \leq k\}$. We then have for $z \neq j$,

$$\mathsf{E}^\lambda\big[\mathsf{E}^\lambda[f(Y)|X]\mathsf{E}^\lambda[f(Y)|X]^\intercal\big] = \mathsf{E}[f(Y_{k+1}^z)f(Y_{k+1}^j)^\intercal] = \mathsf{E}[f(Y_{k+1}^j)f(Y_{k+1}^z)^\intercal]$$

This justifies the choice $\widetilde{m}^{k+1} = \dfrac{1}{Z} \sum\limits_{z=1}^{Z} f(Y_{k+1}^z)$ and

$$\widetilde{\Sigma}^{k+1} = \frac{1}{Z} \sum_{z=1}^{Z} f(Y_{k+1}^z) f(Y_{k+1}^z)^{\mathsf{T}} - \frac{1}{Z^2 - Z} \sum_{z=1}^{Z} \sum_{j \neq z} f(Y_{k+1}^z) f(Y_{k+1}^j)^{\mathsf{T}}$$

### E.2 APPLICATION

We consider a signal tracking problem such as Cammardella et al. (2020). A large population of water heaters is considered during a day, with a time discretization of 15 minutes and thus $T = 96$ points of time discretization. Each water heater $i$ is represented at each time step by $X_t^i := (\theta_t^i, m_t^i)$ with $\theta_i \in \mathbb{R}$ modelizing the average temperature and $m_i \in \{0, 1\}$ modelizing the mode (on/off) of the heater. This is conveniently formulated as an optimal transport problem over the space of distributions on

$$\mathcal{X} = \underbrace{[\theta_{\min}, \theta_{\max}]^T}_{\mathcal{S}} \times \underbrace{[0, 1]^T}_{\mathcal{W}}.$$

If we want to use Algorithm 1, we need at each iteration to compute the gradient on $\mathcal{X} \times \mathcal{W}$, with complexity of $N_\theta{}^T \times 2^{2T}$, with $N_\theta$, being the number of discretization points for the temperature. In this example, this value is thus really high even if $N_\theta$ is low (for example, if $N_\theta = 10$, the complexity is of order $10^{153}$). Therefore Monte Carlo methods are very useful.

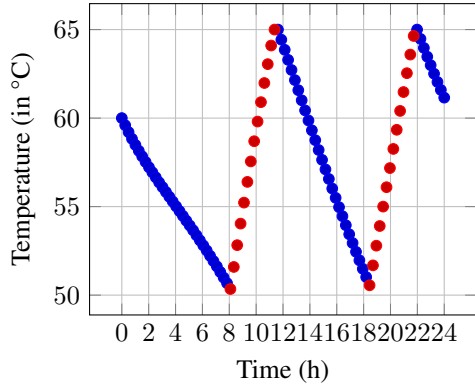

Figure 9: Trajectory of temperature obtained with $\mu_1$: switch when max or min temperatures are reached

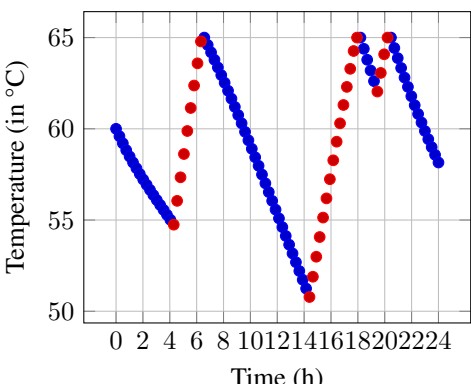

Figure 10: Trajectory of temperature obtained with $\mu_2$: switch at 4am and 7pm and when max or min temperatures are reached

The temperature $\theta_i$ is influenced by a loss effect due to the difference of temperature between the inside and the outside of the tank, which depends on the mean temperatures of the tank and the outside, the heating effect of the water heater and a drain effect which models the draining of hot water by users (showers, dishes etc.). Thus, the ODE driving the evolution of temperature is :

$$\frac{d\theta}{dt} = \underbrace{-\rho(\theta - \theta_{amb})}_{\text{loss effect}} + \underbrace{\sigma m P_{\max}}_{\text{heating effect}} - \underbrace{\tau(\theta - \theta_{in})d(t)}_{\text{drain effect}} \tag{39}$$

As this ODE is deterministic, differences between each distribution of trajectories $\mu$ will be on the mode switch. Most water heaters start to heat when they reach a certain temperature $\theta_{\min}$ and stop heating when they reach a temperature $\theta_{\max}$. This behavior, ensures that the temperature stays in the interval $[\theta_{\min}, \theta_{\max}]$ and will be our $\mu_1$ distribution with one example of trajectory shown in Fig. 9. Our $\mu_2$ distribution considers allowing at most 2 changes of mode compared to the nominal behaviour. This $\mu_2$ behavior also automatically switch modes when arriving to $\theta_{\min}$ or $\theta_{\max}$. One example is shown in Fig. 10. Limiting the number of mode changes has the double advantage of limiting the calculation time and avoiding unrealistic solutions with a high number of mode changes which would not be desirable in practice.

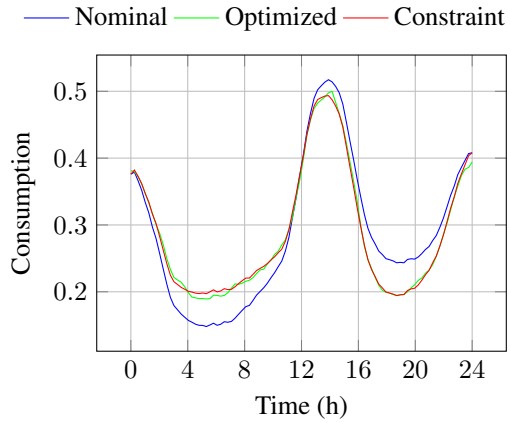

Figure 11: Control of 3000 Water Heaters

The constraint functions $f$ correspond to the aggregated consumption of all water heaters being equal to the reference signal shown in orange in Fig. 11. This signal is a small deformation of the nominal consumption shown in blue when water heater follows the $\mu_1$ distribution. To solve this problem without computing the whole gradient, we use the Split Sampling method proposed in subsection E. With $\varepsilon = 0.01$, $K = 3000$ Water Heaters and $Z = 100$, we obtain the aggregated consumption shown in Fig. 11. We observe that the aggregate consumption tracks well the reference signal, even if there is a slight noise due to the fact that the consumption is only approximated by Monte Carlo methods and not calculated perfectly.

