# OpenReview forum: "Moment Constrained Optimal Transport for Control Applications"
_ICLR.cc/2025/Conference — Submitted to ICLR 2025_

### Official Review · Reviewer_xYHb · 2024-10-31

**Soundness:** 1
**Presentation:** 1
**Contribution:** 2
**Rating:** 3
**Confidence:** 3

**Summary:**

This paper combines the fields of mean field control and optimal transport to solve multi agent control problems. When doing so, the authors constrain some of the marginal distributions to specific moment classes and adopt the Sinkhorn algorithm to their setup. The theoretical and algorithmic considerations are complemented by an extensive example of EV charging in the Netherlands, based on a real dataset.

**Strengths:**

-	The paper tackles the important, contemporary problem of charging EVs and uses a dataset to do so.
-	The combination of mean field control and optimal transport to optimize EV charging appears to be interesting.
-	The paper contains theoretical results to complement empirical findings.

**Weaknesses:**

-	Introduction: In my opinion, the introduction should be less technical in the sense that there shouldn’t be several extensive mathematical expressions. In this way, especially non-expert readers can get a first impression of the paper’s contributions without being confronted with mathematical details.
-	Line 74: EVs are mentioned for the first time here (excluding the abstract). In my opinion the authors should lead with a motivating example and then move on to the technical tools like MFC. In in its current form, the introduction reads like a random collection of mathematical concepts. The authors should put more emphasis on their goals and high-level ideas.
-	Line 84-90: The contributions are formulated far too vague, for example, ‘‘Coordination of an ensemble of agents to achieve a desired goal’’ basically describes any cooperative multi-agent problem. Similarly, a discussion and comparison to the existing literature is missing. What sets the contributions of this paper apart from the existing literature?
-	Assumptions (A1) to (A3): The assumptions are neither explained nor is there a discussion of how realistic or restrictive they are.
-	Proposition 1, Proposition 2, Lemma 1: Like the assumptions, the theoretical results are just stated but not discussed or explained. This presentation style makes it very hard to follow the train of thought in this paper.
-	Section 2.2: It would be helpful for first time readers to explain why the dual problem can be useful for solving these types of problems. Just stating that it is “needed for the algorithm” (line 140) does not provide any intuitive insight.
-	Section 2.3: In this section it is hard for me to understand the algorithmic contributions of the paper. If the contribution, compared to the existing Sinkhorn algorithm, is just the update of $\zeta^k$, the authors should explain when and why this update makes an important difference.
-	Section 3: I am not very familiar with the EV charging literature, but I wonder if there are not any existing papers that focus on similar use cases. Since there is not a single reference in Section 3, it seems like this model is completely new and has no connections to existing work. Is this really the case?
-	Section 4 (like the previous concern): Aren’t there any existing methods to compare against? What exactly are the advantages of the proposed approach?


Minor Comments:

-	Line 35: Is it supposed to be “… common state space $\mathcal{X}$ …’’? How is this state space defined? Are there any restrictions on $\mathcal{X}$ or is it completely arbitrary? (Line 101 seems to contain the precise definition)
-	Line 36: the extensive mathematical definitions should appear later in the paper, but not in first paragraph of the introduction. That aside, I think that $\mu_1$ and $\mu_2$ are not properly defined here.
-	Line 58: What values can the variables $S_k$ and $W_k$ take?
-	Line 73: space missing at “… $S$.It“
-	Lines 79-82: Although the sentence “Inspired by … of optimal control solutions” is somewhat vague, it is nevertheless informative about the goals of this paper. I think it needs to appear earlier in the introduction.
-	Line 103: Are the marginals defined correctly? Shouldn’t the two $dx$ for the first marginal just be $x$? The same question applies to the second marginal and $dy$.
-	Line 104: there is one word “problem” too many.
-	Line 104: How are the probability kernels $T^\lambda$ in this family defined?
-	Line 118: I think the notation is wrong here. It should be “… \leq 0 for all 1 \leq m \leq M}” instead of “… \leq 0: 1 \leq m \leq M}”
-	Lines 119-120: While I do understand that an equality can be equivalently written as two inequalities, I am unsure how this applies to the previously defined moment class. Does that mean that for equalities, we would define a moment class with more than $M$ inequalities?
-	Line 383: Why is a quadratic penalization chosen? If it is standard in the literature, corresponding references should be added.
-	Line 397: Why is the infinite norm a good candidate?
-	Line 419: It should be “(ii)” instead of “(i)”, right?

**Questions:**

Please see the "Weaknesses" section for my questions.

---

> ### Author Response · Authors · 2024-11-20
>
> Another reviewer also said that the contribution section is unclear. Here is a revised version:
>
> Our contributions are the following:
>
> 1) We propose a new problem (MCOT-C) inspired by Optimal Transport and designed to achieve Mean Field Control goals: (i) Agents are controlled to meet a global constraint (ii) Individually, their many strong constraints must be respected, whether physical (an EV cannot be plugged in before it arrives, and its state of charge on arrival or departing time cannot be controlled) or in terms of quality of service (each EV must be fully charged when leaving).
>
> 2) We propose an algorithm to solve it, with a Sinkhorn update on one side and a gradient descend update on the second side.
>
>
> 3) We illustrate this approach with 2 use cases: Respecting grid constraints for charging EVs (Section 3) and tracking a consumption signal for water heaters (Appendix E).
>
> 4) We extend this approach to an online setting and show its efficiency on a case study with a real data set.
>
> In our view, there are several benefits to this approach for the mean-field control literature: (i) It makes the link with the field of optimal transport, and therefore, we are establishing the theoretical background to leverage computational techniques from optimal control theory to the mean field control problem (ii) This new MFC problem introduces a new type of penalization: a Wasserstein cost for the deviation from a policy while enforcing moment constraints. We believe that this penalization is useful and the experiments aim to demonstrate this interest on several case studies. The point of these experiments is to demonstrate that we achieve the MFC goals and to explain how to use this model in practical use cases.
>
> We thank you for pointing out the typos.
>
> Line 58: $S_k$ is an exogenous variable. For the EV in section 3, it is for example $(t_a,b)$, the time of arrival and the state of charge of the battery at the arrival. $W_k$ is the control variable. For the EV in section 3, it is for example $t_c$, the starting charging time.
>
> Lines 119-120: We could define the moment class with $2M$ inequalities. However, in practical implementation, it's simpler to think of them as equalities and have just one $\lambda$ instead of two. The only difference will be in the algorithm with no positive part for these variables.
>
> Line 383: It is standard, we will add the corresponding references.
>
> Line 397: Here we consider the norm of the deviation from the respect of the constraint. In the EV example, it is the gap between aggregate consumption and the constraint. Choosing the infinite norm means that, at each instant, we are within $\varepsilon$ of the constraint we wish to respect. Choosing the 1 or 2 norm, for example, would leave open the possibility of large deviations at certain times. So there's an interesting physical meaning here.

---

> ### Comment · Reviewer_xYHb · 2024-11-26
>
> I thank the authors for their response. While I appreciate their answers, I still think that the overall presentation and structure of the paper require significant improvements. Therefore, I will keep my initial score.

---

### Official Review · Reviewer_Lk5Y · 2024-11-01

**Soundness:** 3
**Presentation:** 3
**Contribution:** 3
**Rating:** 6
**Confidence:** 3

**Summary:**

This is a half-theoretical, half-application paper which initially seeks to propose and solve the moment-constrained optimal transport for control problem, a variation of the constrained optimal transport problem for mean field control. After solving its first objective, the paper seeks to illustrate, and partially adapt, the developed results on an example of charging a large fleet of electric vehicles.

**Strengths:**

- the paper seeks to connect two important areas of applied mathematics, and does it in an elegant fashion that is amenable to an analytical solution
- the paper is largely well-written and I suspect that it would be considered fairly easily readable by experts in the field
- the rigorous approach of the paper is refreshing and exposes the mathematical meat of the problem

**Weaknesses:**

In short, I like this paper a lot, but I am just not convinced that ICLR is the right venue for it. I urge the authors to consider once again whether the ICLR audience is what this paper is shooting for (without any offense at all to either the paper or the ICLR audience). For instance, the L in ICLR stands for "learning". Yet, any connection to learning -- if there is any -- remains unexplained. My issues largely stem from this perceived mismatch:

- relegating the proofs, including the central results (Proposition 1/2), to the appendix probably does not make the paper more readable to the non-experts (who will find notation burdensome to start with), and only signals (possibly incorrectly) that the authors do not see these results as their primary contribution. In that case, I am not sure what is the primary contribution

- there seems to be some dissonance between the claimed contributions (which speak about control of an ensemble of agents), the actual technical contributions (which can indeed serve this goal, but the connection is not really described in detail), and the application (which is farfetched at best: why should the power consumption exactly track a reference trajectory?)

- the "second" technical part of the paper, coming within the application section, applies largely domain-agnostic mathematical theory to a problem that is tailored to the application (which is, as I already mentioned, farfetched). My suggestion would really be to split this paper into two: the first one dealing with the general problem of MCOT-C (and its online version in as much generality as possible), with perhaps only a small academic example if there is no room for more, and the second one applied, with a realistic case study and a detailed description of the algorithm implementation and possible minor modifications

- partly because of decoupling of the proof (and a weird ordering of the proofs in the Appendix), it is not clear to me how challenging the main result actually is: the duality seems rather straightforward. If this is not the case, that should be emphasized. If it is, perhaps it would be good to try to develop online results on MCOT-C in more generality

**Questions:**

I am not sure I have specific questions; perhaps one is "is the control specification in your EV charging case study truly realistic?". My issues are not necessarily fixable by a small change in the paper.

---

> ### Author Response · Authors · 2024-11-20
>
> In this paper, we considered 3 types of control specifications:
>
> - Signal Tracking: The global consumption should follow a certain signal.
>
> - Slope (or gradient) control on the global consumption: The global consumption must not increase or decrease by more than a certain value per unit of time.
>
> - Maximum Power constraints: The global consumption must stay below a certain value.
>
> Your question seems to concern Signal Tracking. To understand the relevance of this topic, it's important to understand that the power grid must be balanced between production and consumption. This balance will be even more complex to achieve in the future, given that renewable energies being developed all over the world are intermittent. To avoid the economic and carbon cost of restarting fossil-fuel power plants to compensate for the intermittent nature of renewable energies, or for peaks in consumption (e.g. plugging in all vehicles coming home from work at 7 p.m.), one possibility being considered is to control a proportion of consumers with flexible consumption patterns. In this context, tracking a certain signal is particularly interesting (\url{https://learn.pjm.com/three-priorities/buying-and-selling-energy/ancillary-services-market/regulation-market#:~:text=The%20Regulation%20D%20signal%20is,the%20system%20need%20for%20regulation.}). In our example, we could imagine the charging station manager having an agreement to consume a predetermined proportion of energy at a given time. This is not a far-fetched example, even if the values are chosen arbitrarily. In fact, this is a research topic for people interested in this type of problem (called Demand Response) such as [1] or [2] for example.
>
> Concerning the other two constraints, they are very useful for a grid manager, whether to avoid exceeding a local power capacity or to avoid excessive peaks.
>
> [1] Adrien Seguret, Optimal control and incentives for decentralized mean field type systems, PSL University, 2023, http://www.theses.fr/2023UPSLD016/document
>
> [2] K. Mukhi and A. Abate, An Exact Characterisation of Flexibility in Populations of Electric Vehicles, 2023 62nd IEEE Conference on Decision and Control (CDC), 2023, pp. 6582-6587, doi: 10.1109/CDC49753.2023.10383521.

---

> > ### Comment · Reviewer_Lk5Y · 2024-11-26
> >
> > I appreciate the authors' reply. In terms of the response itself, I agree with the overall narrative, but imagine that tracking an exact signal might be too restrictive, rather than keeping the consumption/production within a particular "band". In any case, as I mentioned, my question was minor and my concerns are not fixable by any particular clarification. I would like to keep my initial score, and wish the authors all the best with their continued work.

---

### Official Review · Reviewer_1quT · 2024-11-01

**Soundness:** 2
**Presentation:** 1
**Contribution:** 2
**Rating:** 3
**Confidence:** 2

**Summary:**

The authors propose a solution for an application concerning the charging of a fleet of electric vehicles, where a central planner decides on the plugging time of the vehicles. Their proposed method applies a modified version of the Sinkhorn algorithm, typically used in optimal transport problems, to the considered control problem.

**Strengths:**

The authors propose an interesting and relevant control application.
They bridge the fields of optimal transport and mean field control by modifying the Sinkhorn algorithm, where the update on the second marginal is replaced by gradient descent.

**Weaknesses:**

The main weakness of the paper is its structure.
Instead of introducing the considered problem in the introduction, the authors start with their definitions of optimal transport and mean field control. The considered problem is formulated later. This makes it hard for the reader to follow.
When presenting a paper which mainly focuses on one application, it would be better to first clearly describe the application and then introduce the math and methods needed for the solution.

Another major issue is the lack of consistency in the notation. Some examples:
In Eq. 7 there is the variable $l$ which has not been introduced before.
In Eq. 16 there is $h$ on the left hand side and $f$ on the right hand side. Are these equivalent?
In section 3.1 it say the gradient is calculated on $\mathcal X \times \mathcal W$. Should this be $\mathcal S \times \mathcal W$?

Additionally there are many typos, missing punctuation and missing comma placement, further reducing the readability of the paper.

Finally the experiments do not compare to a baseline, apart from a naive decision rule, making it hard to evaluate the efficacy of the proposed method.

**Questions:**

Why is the complexity in section 3.1 $N_t^3 \times N_b$, when the state consists of only two variables with time, the arrival time and the plugging time?

---

> ### Author Response · Authors · 2024-11-20
>
> We thank you for pointing out these typos.
>
> For section 3.1, it is not a typo, the gradient is effectively calculated on $\mathcal{X}\times\mathcal{W}$. In the algorithm (section 2), the gradient is $\sum_{i,j}f_{j}u^k_{i}C_{i,j}e^{{\zeta^k}^{T}f}$, so the gradient should be computed on $\mathcal{X}\times\mathcal{X}$ (i and j are indices of $\mathcal{X}$). But as $\pi$ belongs to $K$ and because of the dirac term $\delta_{x_s}(dy_s)$ in $K$, this sum is simplified as a sum on $\mathcal{X}\times\mathcal{W}$. This also answer your question, in section 3.1, the size of $\mathcal{S}$ is $N_t\times N_b$ and the size of $\mathcal{W}$ is $N_t$, thus the complexity of computing one step is the size of $\mathcal{X}\times\mathcal{W}$ which is $(N_t^2\times N_b)\times N_t $.

---

> > ### Comment · Reviewer_1quT · 2024-11-25
> >
> > I thank the authors for their response.
> > I keep my original score.

---

### Official Review · Reviewer_k6Ue · 2024-11-02

**Soundness:** 2
**Presentation:** 1
**Contribution:** 2
**Rating:** 3
**Confidence:** 2

**Summary:**

This paper introduces moment constrained optimal transport for control (MCOT-C), which leverages computational techniques from optimal control theory for control problems. They provide an algorithm obtained by modifying the Sinkhorn algorithm by replacing the update on the second marginal with gradient descent on the dual. Then, they provide how their proposed approaches apply to mean field control applications, further providing an online version of MCOT-C.

**Strengths:**

The approach of leveraging computational techniques from optimal control theory for control problems and the observations obtained from experiments applying the approach can be interesting. They provide the background theoretical derivation of such an approach. The approach focuses on a finite set of moments, so it could be more tractable in practice.

**Weaknesses:**

**Writing:**
The reviewer thinks the writing of this paper needs to be improved. The reviewer was confused by the abstract and couldn't understand what contributions were made in this paper at first. The authors barely use phrases like 'this paper' or 'we,' so the actions taken in the paper were not distinguished clearly. It seems this issue occurs throughout the paper as well. The reviewer feels that the authors didn't clearly articulate the prior approaches, what they did new, and what the advantages are. This makes it challenging to understand the contributions they are claiming.

**Contribution:**
To the best of the reviewer's understanding, the contribution of this paper is that they are establishing the theoretical background to leverage computational techniques from optimal control theory to the mean field control problem, as presented in sections 2.1 and 2.2. In Proposition 2, they provide the calculation of derivatives for their problem and introduce a gradient descent-based algorithm in section 2.3. Then they directly jump to the experiments, and sections 3 and 4 consist of explanations of their experiments. However, the reviewer believes they should have provided more discussion about the method before proceeding to the experiments, for example, the motivation behind the specific design of the algorithm, or theoretical analysis, or some justifications for why they expect it would work well.

Additionally, the reviewer couldn't grasp what the authors were trying to claim with the experiments. The reviewer believes the authors should have provided guidance on interpreting their experimental results and offered clear conclusions or messages derived from the experiments. However, most of the content in the experiment section seems to focus on the details of the experiments.

**Minor issues:**
- Typo in line 072; a space between "S.It" is required.
- Typo in line 077; "litterature" should be corrected.
- Typo in line 419; "(i)" should be changed to "(ii)."

**Questions:**

- What is $l$ in equation (7)?
- Could you provide what the core observations and messages are that you want to share through the experiment sections?
- Could the authors reorganize the contributions they want to claim in an itemized format? It would be helpful if the reorganized contributions included a clear explanation of the new approaches, the challenges faced, and the advantages compared to previous approaches. Additionally, if the authors feel there are any points that the reviewer may have overlooked, highlighting those would be welcome.

---

> ### Author Response · Authors · 2024-11-20
>
> Our contributions are the following:
>
> 1) We propose a new problem (MCOT-C) inspired by Optimal Transport and designed to achieve Mean Field Control goals: (i) Agents are controlled to meet a global constraint (ii) Individually, their many strong constraints must be respected, whether physical (an EV cannot be plugged in before it arrives, and its charging time and level cannot be controlled on arrival) or in terms of quality of service (each EV must be fully charged before leaving).
>
> 2) We propose an algorithm to solve it, with a Sinkhorn update on one side and a gradient descend update on the second side.
>
> 3) We illustrate this approach with 2 use cases: Respecting grid constraints for charging EVs (Section 3) and tracking a consumption signal for water heaters (Appendix E).
>
> 4) We extend this approach to an online setting and show its efficiency on a case study with a real data set.
>
> In our view, there are several benefits to this approach for the mean-field control literature: (i) It makes the link with the field of optimal transport, and therefore, as you say, we "are establishing the theoretical background to leverage computational techniques from optimal control theory to the mean field control problem," (ii) This new MFC problem introduces a new type of penalization: a Wasserstein cost for the deviation from a policy while enforcing moment constraints. We believe that this penalization is useful and the experiments aim to demonstrate this interest on several case studies. The point of these experiments is to demonstrate that we achieve the MFC goals and to explain how to use this model in practical use cases. These case studies are currently of particular interest to the Mean Field Control community, which is interested in Demand Response methods (https://learn.pjm.com/three-priorities/buying-and-selling-energy/ancillary-services-market/regulation-market#:~:text=The%20Regulation%20D%20signal%20is,the%20system%20need%20for%20regulation. e.g. for Signal Tracking). Using this approach, we can move from a very large problem (the size of the $\mathcal{X}$ space is potentially infinite) to a problem with $M$ the number of constraints.
>
> The definition of $l$ has been misplaced in the appendix (definition of $l_0^\lambda$ in equation (21)) and we will move and correct it.

---

> > ### Comment · Reviewer_k6Ue · 2024-11-26
> >
> > Thank you for addressing my comments so thoroughly. I believe this paper makes a notable contribution. However, I still think that the content needs further development to make it more comprehensive and improvements in the presentation. As other reviewers pointed out, I believe a comparison with prior work is necessary in the case studies. Demonstrating how the proposed algorithm works in specific use cases is meaningful, but I believe the experiment should also convey how it benefits compared to prior approaches. Therefore, I maintain my original score.

---

### Official Review · Reviewer_BBYP · 2024-11-02

**Soundness:** 2
**Presentation:** 1
**Contribution:** 2
**Rating:** 3
**Confidence:** 3

**Summary:**

This paper studies the application of optimal transport to mean-field control. The main contribution is a representation of the mean-field control, Moment Constrained Optimal Transport for Control (MCOT-C), that aims to coordinate the agents and enforce constraints. The authors propose a variant of the Sinkhorn algorithm and apply the proposed algorithm to an EV charging application.

**Strengths:**

The EV Charging problem has received much attention recently. This work aims to optimize the consumption while satisfying the grid constraints.

**Weaknesses:**

The theoretical part of this paper is hard for me to follow. The introduction starts with the mathematical problem settings without sufficient discussion about the background, significant challenges, and the motivation for using optimal transport in mean-field control. Besides, the theoretical problem setting, assumptions, and propositions are difficult to interpret. I suggest the authors add more discussions about the connections between the general framework and a specific example (e.g., the EV charging problem) that is easier to understand. Discussing the intuition/significance after stating each proposition or lemma is also helpful.

For the experiment part, I encourage the authors to compare the problem setting and the performance with related works on EV charging (e.g., [1]). Such comparisons can help the readers understand the advantages/limitations of the proposed approach.

[1] B. Alinia, M. H. Hajiesmaili, Z. J. Lee, N. Crespi, and E. Mallada, "Online EV Scheduling Algorithms for Adaptive Charging Networks with Global Peak Constraints," in IEEE Transactions on Sustainable Computing, vol. 7, no. 3, pp. 537-548, 1 July-Sept. 2022.

**Questions:**

Please see my comments about the weakness.

---

### Official Review · Reviewer_QtSM · 2024-11-04

**Soundness:** 2
**Presentation:** 2
**Contribution:** 2
**Rating:** 3
**Confidence:** 3

**Summary:**

This paper considers using optimal transport (OT) in mean field control, with constraints on the moments of the distributions. An algorithm (Sinkhorn algorithm) is proposed and an example of EV charging is considered.

**Strengths:**

- The problem being considered is an interesting one.
- The idea of controlling an distribution to look like another (more optimal) distribution is certainly useful.

**Weaknesses:**

It's important to note that I'm not that familiar with the field of optimal transport. But I felt the following are the weaknesses:
- The abstract, intro conclusion seem to promise a lot more than what the math actually delivers? The algorithm relies on Gibbs kernels, which feels pretty standard. How broadly applicable is this?
- The EV problem presented is somewhat strange. The paper seems to say that the controllable variables are the EV arrival time and state-of-charge? But these are typically the main source of randomness in EV problems. The controllable knobs are typically the charging profiles.
- How many EVs does there need to be for a mean field approximation to be valid? At any single charging station, there won't be that many EVs.

**Questions:**

- Some comparison against existing EV charging algorithms (there are a lot) would be useful.

---

> ### Author Response · Authors · 2024-11-20
>
> Our algorithm uses Gibbs kernel which are standard in the Optimal Transport literature. In our opinion, our contribution concerning the algorithm is that (i) this algorithm with a sinkhorn update on one side and a gradient descend update on the second side is new in the Optimal Transport literature (ii) this algorithm is fitted for Mean Field Control applications where such methods with Gibbs kernel have not been used to the best of our knowledge. Using this approach, we can move from a very large problem (the size of the space $\mathcal{X}$ is potentially infinite) to a problem with $M$ the number of constraints.
>
> In subsection 3.1, we present the space $\mathcal{W}$ of variables that can be controlled (charging start time) and the space $\mathcal{S}$ of variables that cannot be controlled (EV arrival time and state of charge). This means that we must keep the time and state of charge on arrival, but we can delay when the vehicle start charging.
>
> In section 4, the number of vehicles arriving during the day is around $750$ and the mean field approximation is quite accurate. In the literature, the number of EVs used is from a few hundred to a few thousand ([1] and [2] for example). This number could not be achieved on a single charging station but rather on the scale of a city's charging stations. "Expliquer le cas d'étude typique"
>
> [1] Zejian Zhou, Hao Xu, Mean Field Game-based Decentralized Optimal Charging Control for large-scale of electric Vehicles, IFAC-PapersOnLine,Volume 55, Issue 15,2022, Pages 111-116,ISSN 2405-8963, https://doi.org/10.1016/j.ifacol.2022.07.617.
>
> [2] Muhindo, S.M. Mean Field Game-Based Algorithms for Charging in Solar-Powered Parking Lots and Discharging into Homes a Large Population of Heterogeneous Electric Vehicles. Energies 2024, 17, 2118. https://doi.org/10.3390/en17092118

---

### Meta-Review · Area_Chair_NzZA · 2024-12-17

**Metareview:**

While the reviewers did not find any disqualifying technical errors for the submission, and are appreciative of its general contributions, its connections with machine learning remain tenuous, and the authors have not sufficiently endeavored to make the connection to problems involving distribution matching or constraints that arise in ML problems.

**Additional Comments On Reviewer Discussion:**

The reviewers were mostly concerns about the relevance of the experimental analysis to machine learning problems, as well as broader connections of the technical setting.

---

### Decision · Program_Chairs · 2025-01-22

Reject